



# Sources of nitrous acid (HONO) in the upper boundary layer and lower free troposphere of North China Plain: insights from the Mount Tai Observatory

Ying Jiang[1], Likun Xue[1,4*], Rongrong Gu[1], Mengwei Jia[2], Yingnan Zhang[1], Liang Wen[1], Penggang Zheng[1],

Tianshu Chen[1], Hongyong Li[1], Ye Shan[1], Yong Zhao[3], Zhaoxin Guo[3], Yujian Bi[3], Hengde Liu[3], Aijun

Ding[2,4], Qingzhu Zhang[1], Wenxing Wang[1]

[1]Environment Research Institute, Shandong University, Qingdao, Shandong, 266237, China

[2]School of Atmospheric Sciences, Nanjing University, Nanjing, 210023, China

[3]Taishan National Reference Climatological Station, Tai'an, Shandong, 271000, China

[4]Collaborative innovation Center for climate Change, Jiangsu Province, Nanjing, 210023, China

*Correspondence to: Likun Xue (xuelikun@sdu.edu.cn)

**Abstract:** Nitrous acid (HONO) is a significant precursor of atmospheric "detergent" OH radicals, and plays a vital role in tropospheric chemistry. The current knowledge about the daytime HONO sources is incomplete, and its impact on the tropospheric radical chemistry has not been fully quantified. Existing observational studies of HONO were mostly conducted at surface, with few efforts focusing on the high-elevation atmospheres. In order to better understand the characteristics and sources of HONO in the upper boundary layer and lower free troposphere, two intensive field observations were carried out at the summit of Mt. Tai (1534 m a.s.l.), the peak of the North China Plain, in winter 2017 and spring 2018. HONO showed moderate concentration levels (0.15±0.15 and 0.13±0.15 ppbv), with maximum values of 1.14 and 3.23 ppbv in winter and spring, respectively. Diurnal variation patterns with a broad noontime maximum and lower nighttime concentrations were observed during both campaigns, which is distinct from most of the previous studies at the ground level. The WRF-FLEXPART simulations indicated the combined effects of the planetary boundary layer evolution and valley breeze on the daytime HONO peak. A photostationary state (PSS) analysis suggested the strong unknown daytime HONO source with production rates of 0.45±0.25 ppb/h in winter and 0.64±0.49 ppb/h in spring. Correlation analysis supported the important role of photo-enhanced heterogeneous conversion of $NO_2$ to HONO on the aerosol surface at this high-elevation site. HONO photolysis is the predominant primary source of OH radical and plays a major role in the radical chemistry at Mt. Tai. The model only considering homogenous HONO source would largely underestimate the $HO_x$ radical levels and



atmospheric oxidation capacity in the high-altitude atmosphere. This study sheds light on the characteristics, sources, chemistry, and impacts of HONO in the upper boundary layer and lower free troposphere in the NCP region.

## 1 Introduction

Nitrous acid (HONO) is an important reactive trace gas in the Earth's atmosphere and plays a pivotal role in the tropospheric oxidation chemistry. Photolysis of HONO by sunlight releases the hydroxyl radical (OH), which initiates the oxidation reaction cycles, resulting in the degradation of various primary contaminants and formation of secondary pollutants such as ozone ($O_3$) and secondary aerosols (Alicke et al., 2003). HONO can react with amines to form nitrosamines, which are mutagenic and carcinogenic, and thus affects the human

health (Sleiman et al., 2010). HONO is also a key player in the biosphere-atmosphere interactions and the biogeochemical cycle via dry and wet depositions as well as soil microbial processes (Sörgel et al., 2015). Therefore, it is of high significance to better understand the characteristics, sources, sinks, and environmental consequences of tropospheric HONO.

In comparison to the sink processes that have been relatively well understood, the sources of ambient

HONO are still under extensive exploration and discussions. In the ambient atmosphere, HONO can be either directly emitted from various processes such as combustion and soil emissions (Su et al., 2011; Nie et al., 2015), or formed from the photolysis and chemical reactions of reactive nitrogen-containing substances. The gas-phase reaction of NO with OH (R1) has long been known as a significant formation pathway of HONO (Platt et al., 1980), which forms a pseudo-steady state with the fast photolysis of HONO under sunlight (R2)

and the reactions with OH (R3) during the daytime. The HONO concentrations in photostationary state (PSS; [HONO]$_{PSS}$) can be calculated by Eq. (E1). However, the field observations in the last decade have indicated that the [HONO]$_{PSS}$ can only account for a small portion of the measured concentrations in a variety of environments (Kleffmann et al., 2005; Zhou et al., 2007; Michoud et al., 2014), suggesting the action of additional daytime HONO sources. To explain the observed HONO levels, some new formation mechanisms

have been proposed and tested in lab or through field studies (Bejan et al., 2006; Monge et al., 2010; Su et al., 2011; Li et al., 2014). Among them, a widely well-received HONO source is the heterogeneous reactions of $NO_2$ on various wet surfaces (R4), including ground, buildings, urban grime, and aerosol surfaces (Villena et al., 2011; Wong et al., 2012; Liu et al., 2014; Baergen et al., 2015). However, the contributions of the individual surface types to the ambient HONO formation have not been figured out. Most previous studies suggested the

dominance of ground surface in the heterogeneous HONO formation at the ground level (Alicke et al., 2003; Elshorbany et al., 2009; Ziemba et al., 2010), while some studies argued that aerosol surface may also play an





important role (Colussi et al., 2013; Liu et al., 2014; Tong et al., 2016; Lu et al., 2018).

$$NO + OH \xrightarrow{M} HONO \qquad\qquad (R1)$$

$$HONO \xrightarrow{hv} NO + OH \qquad\qquad (R2)$$

$$HONO + OH \rightarrow NO_2 + H_2O \qquad\qquad (R3)$$

$$[HONO]_{PSS} = \frac{[NO] \times [OH] \times K_{NO+OH}}{J_{HONO} + [OH] \times K_{OH+HONO}} \qquad\qquad (E1)$$

$$2NO_2(g) + H_2O \text{ (surface)} \rightarrow HONO(g) + HNO_3 \text{ (surface)} \qquad\qquad (R4)$$

Currently, the field observational studies of HONO were mainly conducted at the surface level, with few efforts made in the high elevations such as upper planetary boundary layer (PBL) and lower free troposphere (FT). This hindered a thorough understanding of the characteristics, processes, and environmental impacts of tropospheric HONO. First, it can be easily expected that the HONO concentrations should be rather low in
the upper PBL and lower FT, given the short lifetime of HONO under sunlight. This has been confirmed by the limited measurement studies at mountaintop and aboard aircraft (Acker et al., 2006; Zhou et al., 2007; Ye et al., 2016; Ye et al., 2018). Field observations at high-elevation sites could provide direct constraints for better understanding the vertical distribution and transport processes of HONO. Second, upper PBL and lower
FT are free from the terrestrial (especially ground) surfaces and hence provide an ideal opportunity to evaluate the role of aerosol particles in the heterogeneous formation of HONO. Third, the impacts of HONO on the atmospheric oxidation capacity (AOC) in the upper PBL and lower FT have not been fully quantified. Existing modelling studies may underestimate the AOC of high-altitude atmospheres owing to the lack of observational data constraints (Tan et al., 2001). Consequently, in-situ observations of HONO in the upper PBL and lower
FT are very important to achieve a comprehensive understanding of the tropospheric oxidation chemistry.

To investigate the chemical and physical processes in the upper PBL and lower FT over the North China Plain (NCP), one of the most urbanized and polluted regions in China, two intensive field campaigns were carried out at Mt. Tai, the highest mountain over the region, in winter 2017 and spring 2018. This is a part of the Integrated Measurement Campaigns of Air Pollution Complex over the coastal regions of East China
(iMAPEC), that aimed to deploy multiple advanced measurement platforms to investigate the three-dimensional distributions and processes governing the formation of regional air pollution over eastern China. A large suite of chemical species and physical properties were measured during the campaigns. This paper describes the measurement results of HONO and related parameters, which are the first report of HONO





chemistry in the upper PBL and lower FT in the NCP region. Several models including WRF-FLEXPART, PSS calculation, and Master Chemical Mechanism (MCM) box model were used to illustrate the origins, formation, and impacts of HONO. Overall, this study demonstrates the moderate pollution levels of HONO in the upper PBL and lower FT in the NCP, and provides some new insights into the roles of aerosol surface

in the HONO formation as well as the significant effects of HONO photolysis on the atmospheric oxidation capacity in the high-altitude atmospheres.

## 2    Methodology

### 2.1 Site description

The Mt. Tai Observatory has been widely deployed as the sampling platform to investigate the regional

air pollution in the North China Plain region in the past decade (e.g., Gao et al., 2005; Sun et al., 2016; Wen et al., 2018). It is built at the summit of Mount Tai (36°16′ N, 117°6′ E, 1534 m a.s.l.), which is at the center of the NCP (see Fig. 1). The geographical location and altitude of Mount Tai make it representative of the regional background air of the region, and it is well within the upper planetary boundary layer during daytime on sunny days and in the residual layer or free troposphere at night. There are some local emissions from small

restaurants and temples at the mountain, but the station is situated in the less frequently visited eastern part of the summit and thus the impact of local anthropogenic emissions can be neglected (Gao et al., 2005). The city of Tai'an (population: over 5.5 million) is located at the foot of Mt. Tai, about 15 km to the south, and the capital city of Shandong province, Ji'nan (population: over 7.4 million), is situated ~60 km to the north. It should be noted that a 1-yr continuous measurement campaign had been conducted at an urban site of Ji'nan

(Li et al., 2018a), and their results are compared here with those at Mt. Tai to infer the vertical distributions of HONO in the NCP region. Details of the Mt. Tai Observatory can be found in Gao et al. (2005), Zhou et al. (2010), and Sun et al. (2016). Two intensive field measurement campaigns were conducted during 1st-31st December in 2017 and from 5th March to 8th April in 2018.

### 2.2 Measurement techniques

A large suite of chemical, physical, and meteorological parameters were measured in real-time during the campaigns. Here we briefly describe the measurement techniques for the species that were used in the present study. HONO was measured by a long path absorption photometer (LOPAP-03, *QUMA GmbH, Germany*). It detects HONO based on the wet chemistry theory, and two channels are designed to eliminate the possible interferences. The detailed description of the LOPAP instrument has been provided by Heland et

al. (2001), and it has been successfully applied in various environments (Kleffmann et al., 2005; Li et al., 2018a; Wen et al., 2019; Liu et al., 2019). During the present study, manual calibrations were performed using





1000 mg/L nitrite standard solutions every four days to examine the sensitivity of the detector, and the small drift of the baseline was corrected by regular automatic flushing with ultrapure nitrogen (purity of 99.999%), which was done for 30 min at a time interval of 11 h 30 min. The detection limit is 3 ppt at a time resolution of 30 s, and the measurement precision and accuracy are 1% and 10%, respectively.

Nitric oxide (NO) and total nitrogen oxides ($NO_y$) were measured by a commercial instrument (*Model T200U, Advanced Pollution Instrumentation (API), USA*), equipped with an externally placed molybdenum oxide catalytic converter. Nitrogen dioxide ($NO_2$) was monitored by a cavity-attenuated phase-shift spectroscopy that is highly selective for true $NO_2$ (*Model T500U, API, USA*). $O_3$ was measured by an UV photometric analyzer (*Model T400, API, USA*). CO was measured by a gas filter correlation, non-dispersive

infrared analyzer (*Model 300EU, API, USA*). $SO_2$ was measured by an ultraviolet fluorescence analyzer (*Model 43C, Thermo Electron Corporation, USA*). The fine particle ($PM_{2.5}$) mass concentration was measured by a SHARP monitor (*Thermo Fisher Scientific, USA*). Inorganic water-soluble ions (i.e., $NO_3^-$, $SO_4^{2+}$, $NH_4^+$, etc.) in $PM_{2.5}$ together with acid and alkaline gases (i.e., $HNO_3$, $NH_3$, etc.) were detected by a Monitor for AeRosols and GAses (*MARGA, ADI20801, Applikon-ECN, Netherlands*). The particle number size

distributions (5-10000 nm) were measured by a Wide-Range Particle Spectrometer (*WPS, Model 1000XP, MSP Corporation, USA*), which were used to calculate the aerosol surface area density with an assumption that the particles are of spherical shape. Methane and $C_2$-$C_{10}$ non-methane hydrocarbons (NMHCs) were measured by taking ambient air samples in stainless steel canisters on selected days, followed by chemical analysis by GC-FID/ECD/MS in the laboratory of University of California at Irvine (Simpson et al., 2010).

$C_1$-$C_8$ carbonyl compounds were observed offline by absorbing ambient air into *2,4-DNPH* sorbent cartridge followed by HPLC analysis (Yang et al., 2018). Meteorological data including temperature, relative humidity (RH), pressure, and wind speed and direction were provided by Taishan National Reference Climatological Station. Photolysis frequency of $NO_2$ ($J(NO_2)$) was monitored with a filter radiometer (*Meteorologie Consult GmbH, Germany*), and the measurements were only available during the spring campaign. All of these

techniques have been successfully used in many previous studies, where details of the operation, quality assurance and quality control procedures can be found (e.g., Li et al., 2018a; Wen et al., 2018).

    The photolysis frequencies of $J(NO_2)$, $J(HONO)$, $J(O^1D)$ were computed with the NCAR Tropospheric Ultraviolet and Visible (TUV) Transfer Model (http://www.acd.ucar.edu/TUV), with the $O_3$ column density obtained from the Total Ozone Mapping Spectrometer (TOMS, data available at

http://toms.gsfc.nasa.gov/teacher/ozone). In spring when the measured $J(NO_2)$ was available, the calculated $J(HONO)$ and $J(O^1D)$ were scaled with the ratio of measured $J(NO_2)$ to calculated $J(NO_2)$. In winter, without direct observations of $J(NO_2)$, the TUV-calculated J values were approximated by multiplying the average



ratio of measured J(NO$_2$) to calculated J(NO$_2$) obtained in spring (i.e., 0.7).

## 2.3 Lagrangian Particle Dispersion Model

A Lagrangian Particle Dispersion Model (WRF-FLEXPART v3.3) was utilized to investigate the provenance and trajectory of the air masses sampled at Mt. Tai. The Weather Research and Forecasting (WRF) Model, driven by the FNL reanalysis data, was run to produce the high spatial resolution meteorological field. The horizontal resolution of the WRF model simulations was 20 km, and the vertical direction was divided into 30 layers under 100 hPa, including 18 layers below the altitude of 2 km. In the LPDM simulations, 3000 particles were released for 3-days backward trajectories from the observational station at the altitude of 1400 m a.g.l. The domain was in the range of 105°E-126°E and 25.5°N-43.5°N with 0.1°× 0.1° grids. The residence time of particles for a thickness 100 m above the surface in all grid cell was calculated as the "footprint" retroplume, which indicated the distribution of probability or residence time of a simulated air mass at the surface level. This method has been widely applied in the previous studies (Stohl et al., 2003; Ding et al., 2009; Pan et al., 2014).

## 2.4 Chemical box model

A chemical box model was used to simulate the in-situ tropospheric oxidation chemistry and quantify the contributions of HONO to the OH source and atmospheric oxidative capacity. It was set up based on the Master Chemical Mechanisms (MCM, v3.3.1), a nearly explicit chemical mechanism describing the degradation of 143 primary VOC compounds in addition to the latest inorganic reactions (Jenkin et al., 2003; Saunders et al., 2003). The model was constrained with the measured concentrations of NO, NO$_2$, CO, SO$_2$, O$_3$, VOCs and carbonyls as well as the meteorological conditions at a time interval of 5 min. For VOCs and carbonyls for which the measurements were not in real-time, the high-resolution data were approximated based on the CO (temperature) data and the measured correlations with CO (temperature) for anthropogenic (biogenic) VOCs. Such approximation may be subject to some uncertainties but should not significantly interfere the estimation of the role of HONO photolysis in OH sources (Yang et al., 2018). The model computed the major primary production rates of RO$_x$ radicals (RO$_x$=OH+HO$_2$+RO$_2$), including photolysis of O$_3$, HONO, HCHO and other OVOCs, and the ozonolysis reactions of unsaturated VOCs (Xue et al., 2016). Also calculated by the model was the AOC by OH, which is defined here as the reaction rate of OH with NO, NO$_2$, SO$_2$, CO and VOCs (Xue et al., 2016). The model simulations were performed for the two campaigns, and a four-day pre-run was made with constraints of campaign-average data to facilitate the model to a chemical steady state for the unconstrained species. Two model scenarios with and without the measured HONO constraints were done to assess the contribution of the unknown HONO sources to the OH chemistry



(note that the model only contained the homogeneous source of HONO). This model has been extensively adopted to simulate the radical chemistry and ozone formation in our previous studies (Xue et al., 2014; Xue et al., 2016; Yang et al., 2018).

## 3 Results and Discussion

### 3.1 Concentration levels and temporal variations

Table 1 summarizes the statistics of major trace gases, $PM_{2.5}$, and meteorological parameters observed at Mt. Tai during the two campaigns, and the detailed time series of HONO and related species are shown in Figs. S1-S2. The atmospheric conditions at Mt. Tai were featured by a cold and dry weather (especially in winter with average (±standard deviation; SD) temperature and RH of -4.6±3.7 °C and 46±20%) as well as relatively lower concentrations of air pollutants. This was due to the high elevation of the station, which is primarily located in the lower FT and/or upper PBL during the measurement periods. The sampled air masses at Mt. Tai were rather chemically aged, as indicative of the low $NO_x/NO_y$ ratios with average values (±SD) of 0.45±0.21 and 0.25±0.19 in winter and spring, respectively. The average concentrations of $NO_x$ (4.70±4.40 ppbv), $SO_2$ (3.7±3.9 ppbv), and $PM_{2.5}$ (40.3±27.5 µg/m³) in winter were significantly higher than those in spring (2.18±2.09 ppbv for $NO_x$, 1.6±1.6 ppbv for $SO_2$, and 33.7±26.7 µg/m³ for $PM_{2.5}$), while an opposite seasonal pattern was found for $O_3$ (48±12 ppbv in winter vs 63±14 ppbv in spring). The ambient levels of HONO were comparable in both seasons, with average mixing ratios (±SD) of 0.15±0.15 and 0.13±0.15 ppbv in winter and spring. Despite the relatively low concentration levels, spikes in HONO (and $NO_x$, $NO_y$, CO and $PM_{2.5}$) concentrations were frequently encountered throughout the measurement periods (see Figs. S1-S2), indicating the transport of polluted plumes to the mountaintop. The maximum HONO mixing ratios were recorded at 1.14 and 3.23 ppbv in winter and spring, respectively. The above inspection of data reveals the higher than expected concentration levels of HONO in the upper PBL and lower FT of the NCP region.

Table 2 compares the HONO concentration levels at Mt. Tai with those measured at various surface sites as well as in the high-elevation atmospheres around the world. Obviously, the HONO concentrations at Mt. Tai were significantly lower than those observed at the polluted surface sites in China. For example, the ambient HONO levels at Mt. Tai were approximately an order of magnitude lower than those at urban or suburban sites in the NCP and Pearl River Delta (PRD) regions. This is within expectation considering the height of Mt. Tai and the short lifetime of HONO. A detailed comparison against the measurement results at a nearby urban site will be made in Section 3.2 to infer the vertical distribution of HONO in this region. Despite the lower absolute levels, a noteworthy result is the $HONO/NO_2$ ratios at Mt. Tai (0.059±0.091 in winter and 0.072±0.080 in spring) are well within the reported ranges (0.041-0.093) obtained at these surface sites (Su et





al., 2008b; Li et al., 2012; Wang et al., 2017; Li et al., 2018a; Liu et al., 2019), implying that $NO_2$ might be the key factor governing the regional distribution (or formation) of HONO. Another important finding from the comparison is that the HONO concentrations at Mt. Tai are substantially higher than those measured from the high-elevation atmospheres, e.g., Whiteface mountain, U.S. (Zhou et al., 2007), Summit, Greenland (Dibb et al., 2002), Hohenpeissenberg, Germany (Acker et al., 2006), Mt. Brocken, Germany (Acker et al., 2001), and aboard research aircraft over the North Atlantic Ocean (Ye et al., 2016), northern Michigan (Zhang et al., 2009), and southeastern U.S. (Ye et al., 2018). This demonstrates the higher pollution levels of HONO in the upper PBL and lower FT in the NCP region, which may further lead to stronger supply of OH radicals and higher atmospheric oxidation capacity in the high-elevation atmosphere. It is very important to pin down the sources of the elevated HONO and fully evaluate their impacts on OH radicals and AOC in the upper PBL and lower FT of the NCP region, which will be detailed in the following sections.

Figure 2 shows the average diurnal variation patterns of HONO and related parameters during the two measurement campaigns. Overall, most pollutants including $NO_2$, $NO_y$, CO, $O_3$ and $PM_{2.5}$ showed well-defined diurnal profiles with a daytime concentration peak compared to lower levels during the nighttime (note that the exact peak time varied among the individual pollutants with different lifetimes and photochemical behaviors). This can be explained by the evolution of the PBL, mountain-valley breeze, and atmospheric photochemistry (Sun et al., 2016). Similar diurnal patterns were also found for HONO, showing noontime concentration peaks with relatively lower mixing ratios during the night in both seasons. Such pattern is quite different from those determined by most previous studies at the ground level, which generally show higher nighttime concentrations but with a daytime trough (Lee et al., 2016; Li et al., 2018a; Liu et al., 2019; Fu et al., 2019). The noontime HONO maximum at Mt. Tai suggested the potential upslope transport of HONO to the mountaintop and/or the presence of 'additional' daytime sources. We also examined the diurnal variations of $HONO/NO_2$, the ratio commonly used to indicate the efficiency of HONO formation from the reactions involving $NO_2$ (Yu et al., 2009). From Fig. 2c, the $HONO/NO_2$ ratio shows an increasing trend throughout the nighttime together with an extra peak during the midday in both seasons. This indicates the nighttime HONO formation and accumulation from the heterogeneous reactions of $NO_2$ as well as the presence of some 'additional' daytime sources. The higher daytime $HONO/NO_2$ ratios in spring than in winter suggest the action of photo-enhanced heterogeneous formation of HONO. The daytime HONO sources will be diagnosed via a detailed PSS calculation in Section 3.3.

## 3.2 Potential transport by PBL evolution and mountain-valley breeze

In this section, we compare our observations at Mt. Tai with the recent year-round measurement results



obtained at a surface site in Ji'nan to infer the vertical gradient of HONO over the NCP region, and examine the vertical transport of HONO with the LPDM modeling. The surface site is located in the downtown of Ji'nan city, which is about 60 km to the north of Mt. Tai (Fig. 1), and the measurement data have been reported by Li et al. (2018a). It should be noticed that both measurements were not conducted at the same time, and only the average conditions of HONO and $NO_2$ in the same season were compared here.

Figure 3 presents the average diurnal variations of HONO, $NO_2$, and HONO/$NO_2$ at both surface and mountaintop in winter and spring. Several interesting aspects are clearly illustrated from the figure. First, the diurnal variation patterns are distinct between at surface and at the mountaintop. HONO showed higher concentrations at nighttime and lower levels during the day at surface, which is opposite to the profile at the mountaintop. Second, the HONO and $NO_2$ concentrations at surface were overall almost an order of magnitude higher than those at Mt. Tai, but the vertical gradient (as indicative of the Mountaintop/Surface (M/S) concentration ratio) showed a clear diurnal dependence. The M/S ratios of HONO at nighttime (18:00-6:00 LT; with averages of 0.05±0.01 and 0.07±0.02 in winter and spring, respectively) were significantly lower than those during the midday (11:00-15:00 LT; 0.28±0.04 and 0.23±0.03). This is also the case for $NO_2$, and should be ascribed to the efficient mixing within the PBL and upslope transport driven by the valley breeze during the daytime. Third, although the HONO/$NO_2$ ratios were almost comparable at both surface and Mt. Tai during the night, the daytime ratios were significantly higher at the mountaintop (0.065±0.093 and 0.093±0.094) than at the surface (0.047±0.090 and 0.052±0.040). It can be argued that the heterogeneous formation of HONO should be stronger at the mountaintop, which may be due to the more intense solar radiation at the high altitudes.

Can the PBL mixing and mountain-valley breeze transport HONO from the ground level to the mountaintop with an altitude of ~1.5 km, given its quite short lifetime? The vertical transport distance ($\sigma$) driven by the turbulent diffusion can be estimated by the following equation (E2) [Jacob, 1999].

$$\sigma = (2K_Z\tau_{HONO})^{1/2} \qquad (E2)$$

Where, $K_z$ is the turbulent diffusion coefficient, and $\tau_{HONO}$ is the chemical lifetime of HONO. In the early morning when the boundary layer is stable, $K_z$ is usually within the range of $10^2 \sim 10^5$ cm$^2$ s$^{-1}$ and increases with the temperature (Zhang et al., 2009). Assuming an overall $\tau_{HONO}$ of 60 minutes in the early morning (corresponding to a J(HONO) of $2.8 \times 10^{-4}$ s$^{-1}$), the vertical transport distance of the ground-level HONO is estimated in the range of 9-268 m, far below the altitude of Mt. Tai. This explains the huge difference in the HONO concentrations between the surface site and the mountaintop in the early morning and during the nighttime. Around noontime, the PBL has been developed and $K_z$ is generally in the order of $2-8 \times 10^6$ cm$^2$ s$^{-1}$



(Donahue et al., 1973). Adopting the commonly used values of Kz ($5\times10^6$ cm$^2$ s$^{-1}$) and $\tau_{HONO}$ (25 minutes; refer to Section 3.3 for the lifetime estimation), the maximum transport distance of HONO can be up to 1200 m, which is still below the peak of Mt. Tai. If we further consider the mountain-valley breeze, by adopting a mean upslope wind speed of 1 m/s (Kalthoff et al., 2000; Wang et al., 2015), the transport height of HONO by the valley breeze is estimated to be 1500 m. Note that the transported HONO from the ground level may be subject to extensive dilution and more photolysis at higher altitudes. Therefore, the PBL evolution and valley breeze should play an important role in the regional transport of HONO in the mountainous areas.

The LPDM modelling results also support the upslope transport of surface pollution to the Mt. Tai. Figure 4 shows the average distributions of the "footprint" retroplume of the three-day backward trajectories, computed by the WRF-FLEXPART model, arriving at Mt. Tai during both daytime and nighttime. During the night and early morning (2:00-8:00 LT), obviously, the air masses sampled at Mt. Tai were mainly originated from long-range transport and less affected by the PBL air from the NCP region (Fig. 4a and 4c). This can be explained by that Mt. Tai is in the free troposphere (or residual layer) and under the influence of downslope mountain breeze during the nighttime. In comparison, the air masses reaching Mt. Tai in the afternoon (12:00-18:00 LT) primarily came from the boundary layer in the nearby regions (i.e., Ji'nan, Tai'an and Ji'ning, major cities surrounding Mt. Tai; Figs 4b and 4d). Note that the "footprint" retroplume denotes the residence time of particles within the lowest 100 m near surface. This demonstrates the efficient transport of regional surface air pollution to the mountaintop with the development of PBL and valley breeze.

### 3.3 Daytime HONO sources

Exploration of the potential daytime source has been a major objective of the recent HONO observational studies. Here we diagnose the possible daytime sources of HONO in the upper PBL and lower FT by performing a PSS analysis for the measurement data collected at Mt. Tai. According to the measurement-derived J(HONO) (i.e., $6.4\pm3.5$ and $9.5\pm3.2\times10^{-4}$ s$^{-1}$ in winter and spring; see Fig. S3) and the model-simulated OH concentrations (i.e., $2.5\pm0.7$ and $4.4\pm2.0\times10^6$ molecules cm$^{-3}$; see also Fig. S3), the average lifetime of HONO was estimated as $25.7\pm1.4$ and $21.8\pm16.9$ minutes during noontime (11:00-14:00 LT) in winter and spring, respectively. Give such short lifetimes, the air masses arriving at Mt. Tai at noon should facilitate a pseudo-steady state for HONO. The PSS concentrations of HONO ([HONO]$_{pss}$) can be deduced from Equation (E1), if we only took the homogeneous formation from OH+NO reactions, HONO photolysis, and OH+HONO reactions into account. Note that the OH concentrations were simulated with the observation-constrained MCM model as described in Section 2.4 (see Fig. S3 for detailed time series of OH). The PSS approach has been widely used in the previous studies (Zhou et al., 2002; Kleffmann et al., 2005; Elshorbany et al., 2012).



Figure 5 shows the campaign-average $[HONO]_{pss}$ at noontime and the comparison with the measured HONO concentrations in the two seasons. It is evident that the $[HONO]_{pss}$ was significantly lower than the measured data. On average, $[HONO]_{pss}$ only accounted for 18% and 12% of the ambient HONO levels at Mt. Tai in winter and spring, demonstrating the existence of strong HONO sources other than the homogeneous

OH+NO reactions. To maintain the observed HONO levels, an additional 'source' is needed and its source strength (hereafter referred to as $P_{other}$) can be calculated from Equation (E3).

$$P_{other} = [HONO]_{obs} \times J(HONO) + [HONO]_{obs} \times [OH] \times K_{OH+HONO} - [NO] \times [OH] \times K_{OH+NO} \quad (E3)$$

In the present study, the $P_{other}$ was calculated in the range of 0.01-1.24 and 0.01-4.26 ppb/h, with average (±SD) values of 0.45±0.25 and 0.64±0.49 ppb/h in winter and spring, respectively. In comparison, the noontime

HONO production rates from the OH+NO reactions were 0.13±0.10 and 0.09±0.08 ppb/h, much lower than $P_{other}$. The additional source strengths of daytime HONO at Mt. Tai are much lower than those previously reported at the surface sites in Ji'nan (2.95 ppb/h; Li et al., 2018a), Beijing (1.83 ppb/h; Hou et al., 2016), and Santiago, Chile (1.69 ppb/h; Elshorbany et al., 2009), and are comparable to those derived from the rural or mountain sites, such as Hohenpeissenberg (0.40 ppb/h; Acker et al., 2006), Whiteface mountain (0.38 ppb/h;

Zhou et al., 2007), and a forest site in Julich, Germany (0.50 ppb/h; Kleffmann et al., 2005). Some studies have reported much lower $P_{other}$ values obtained from aircraft studies in Michigan (0.057 ppb/h; Zhang et al., 2009) and in Southern US (0.17 ppb/h; Ye et al., 2018).

Recent field and laboratory studies have proposed some potential sources that may help explain the observed higher-than-PSS daytime HONO levels. Among them, direct emissions from vehicle exhaust and

soil microbiological processes, heterogeneous conversion of $NO_2$ to HONO on surfaces, and photolysis of adsorbed and aqueous-phase nitrate are relatively well received to be relevant in the ambient atmospheres (Zhou et al., 2003; Su et al., 2011; VandenBoer et al., 2013; Liu et al., 2019). It is believed that vehicle and soil emissions should be of minor importance at Mt. Tai, which is free from anthropogenic activities and soil surfaces. We then diagnosed the potential action of heterogeneous formation of HONO on both ground and

aerosol surfaces as well as photolysis of nitrate, by examining the relationship between $P_{other}$ and a variety of indicators for the specific HONO sources. Such correlation analysis method has been widely applied in many previous studies to identify the most likely additional daytime HONO source, despite the inherent uncertainty in the statistical approach (Su et al., 2008b; Lee et al., 2016; Lu et al., 2018).

Figure 6 shows the scatter plots of $P_{other}$ against several HONO source indicators during the two

campaigns. As illustrated from the figure, $P_{other}$ shows weak correlation with $NO_2$ (the commonly used indicator for the heterogeneous $NO_2$-to-HONO conversion on the ground surface, by assuming that the ground





surface density is uniform in the PBL), with correlation coefficients (r) of 0.26 and 0.24 in winter and spring, respectively. This indicates that the heterogeneous reactions on the ground should not be the dominant HONO source at Mt. Tai. If the aerosol surface area density ($(S/V)_a$) was included, the correlations were significantly improved (Figs. 6b and 6e), with r values of 0.52 and 0.43 between $P_{other}$ and $NO_2*(S/V)_a$ (an indicator of the

heterogeneous $NO_2$-to-HONO conversion on aerosol surface). Furthermore, the correlations were further improved after $J(NO_2)$ (an indicator of sunlight) was considered, with the r being 0.54 and 0.48 for $P_{other}$ versus $J(NO_2)* NO_2*(S/V)_a$ (Figs. 6c and 6f). These results suggest that the photo-enhanced heterogeneous reactions of $NO_2$ on the aerosol surface should be a significant daytime HONO source at Mt. Tai. This is different from most previous studies conducted at the ground level showing the dominant role of ground surface in the

heterogeneous HONO formation (Kleffmann et al., 2003; Su et al., 2008a; Xu et al., 2015; Li et al., 2018a), and highlights the importance of aerosol surface in the HONO formation in the upper PBL and lower FT. Besides, Fig. S4 shows the relatively weak correlation between $P_{other}$ and $J(NO_2)*pNO_3^-$ (the indicator for the photolysis of particulate nitrate), with r of 0.17 and 0.03, indicating the nitrate photolysis is relatively insignificant to the ambient HONO at Mt. Tai.

**3.4 Impacts of HONO on OH radical and atmospheric oxidation capacity**

The detailed chemical budget of $RO_x$ radicals was dissected by the observation-based MCM box model. Figure 7 shows the average diurnal profiles of primary production rates of OH, $HO_2$ and $RO_2$ radicals from major sources at Mt. Tai during the two campaigns, and the statistical results are documented in Table 3. Two points are noteworthy here. First, the primary production rates of $RO_x$ radicals at Mt. Tai were substantially

lower than those derived from most of the ground-level sites over China, such as Beijing (Liu et al., 2012; Yang et al., 2018), Wuhan (Lu et al., 2017), Chengdu (Tan et al., 2018), Hong Kong (Xue et al., 2016; Li et al., 2018b), and a rural site in the Yellow River Delta region (Chen et al., 2020). Note that most of these previous studies were based on modelling of the polluted photochemical episodes, but the modelled highest $RO_x$ production rates at Mt. Tai were still lower than these studies. The model-simulated daily maximum AOC

(defined here as the sum of reaction rates of OH with $NO_x$, $SO_2$, CO and VOCs) at Mt. Tai was in the range of $1.0$-$3.3 \times 10^7$ and $0.4$-$9.9 \times 10^7$ molecules cm$^{-3}$ s$^{-1}$ in winter and spring (see Fig. S5), respectively, which are still much lower than those at the abovementioned surface sites (Xue et al., 2016; Yang et al., 2018; Li et al., 2018b). This demonstrates the relatively lower oxidation capacity of the upper PBL and lower FT atmosphere compared to the polluted surface-layer air. Nonetheless, the primary $RO_x$ production rates at Mt. Tai were

comparable to those obtained from Hohenpeissenberg, Germany (Acker et al., 2006), Whiteface mountain (Zhou et al., 2007), and Pinnacle State Park, a rural site in southwestern New York (Zhou et al., 2002).



Second, photolysis of HONO presents the predominant primary OH source, and is also among the most important sources of $RO_x$ at Mt. Tai. In winter, HONO photolysis was the overwhelming primary source of $RO_x$ radicals, with the mid-day (9:00-15:00 LT) average production rate 0.520 ppb/h, followed by photolysis of OVOCs other than formaldehyde (0.210 ppb/h for $HO_2$ and 0.182 ppb/h for $RO_2$) and formaldehyde (0.193 ppb/h for $HO_2$), ozonolysis reactions of unsaturated VOCs (0.036 ppb/h for $RO_x$), and $O_3$ photolysis (0.022 ppb/h for OH). In spring, in comparison, photolysis of OVOCs other than formaldehyde turns over to be the dominant $RO_x$ source (with mid-day average rate of 0.434 ppb/h for $HO_2$ and 0.407 ppb/h for $RO_2$), then photolysis of HONO (0.511 ppb/h for OH), formaldehyde (0.374 ppb/h for $HO_2$) and $O_3$ (0.178 ppb/h for OH), and reactions of $O_3$+VOCs (0.057 ppb/h for $RO_x$). In percentage, HONO photolysis accounted for 44.4% and 25.8% of the total primary $RO_x$ production at mid-day at Mt. Tai in winter and spring, respectively. Evidently, HONO photolysis plays a very important role in the OH supply and hence jump-start of atmospheric oxidation chemistry in the upper PBL and lower FT in the NCP region.

How much do the additional HONO sources (other than the homogeneous OH+NO source) affect the atmospheric photochemistry in the upper PBL and lower FT? To evaluate this issue, two modelling scenarios were performed, with and without constraints of the measured HONO data. The model only contains the homogeneous HONO formation from OH+NO reaction, and thus the difference between both scenarios can represent the effects of the additional daytime HONO sources. Figure 8 documents the differences in the model-simulated OH, $HO_2$, AOC, and $P_{OH}$ (primary production rate of OH) between the two scenarios in winter and spring. Clearly, the model only considering the homogeneous source and without observational constraints largely underestimated the $HO_x$ radical levels and AOC at Mt. Tai. Specifically, the underestimation in the mid-day (9:00-15:00) average $P_{OH}$, OH, $HO_2$, and AOC can be up to 83.4% (63.7%), 47.2% (27.1%), 39.7% (20.3%), and 46.7% (27.7%) in winter (spring), compared to the base scenario with constraints of the measured HONO data. Given the essential role of $HO_x$ radicals in the atmospheric chemistry, the models would largely underestimate the atmospheric oxidation processes in the upper PBL and lower FT, if the measurement data and/or additional HONO source mechanisms were absence. So far, direct observations of HONO in the high-elevation atmospheres are still lacking, and accurate representation of the HONO sources in atmospheric chemistry models is underway. This study highlights the significance of HONO chemistry in the upper PBL and lower FT in the NCP region, and calls on more efforts to address the tempo-spatial variations and sources of HONO in the high-elevation atmospheres.

## 4. Conclusions

We analyzed the characteristics, sources, and impacts on tropospheric oxidation chemistry of ambient




HONO in the upper PBL and lower FT of the NCP region, by conducting intensive field observations at Mt. Tai (the peak of the region) followed by detailed modelling analyses. Moderate concentration levels of HONO were observed, which are higher than those measured from other high-elevation areas around the world. HONO showed a distinctive diurnal variation pattern at Mt. Tai with a daytime concentration peak compared to lower levels at night, which is opposite to the results determined at the ground level. The evolution of PBL and the upslope transport driven by valley breezes explained the observed daytime HONO peak. The data were compared with the year-round continuous measurement results at a nearby surface urban site to infer the vertical gradient of HONO in the NCP region. Strong additional daytime HONO sources other than homogeneous OH+NO reactions existed in the PBL, and the PSS estimation only accounted for 18% and 12% of the observed HONO in winter and spring. The photo-enhanced heterogeneous reactions of $NO_2$ on aerosol surface appears to be an important HONO source. HONO photolysis is the predominant primary source of OH radical, and plays a major role in the radical chemistry at Mt. Tai. With only inclusion of the OH+NO reactions, the model significantly underestimated the OH (by ~47.2%; 27.1%), $HO_2$ (by ~39.7%; 20.3%), $P_{OH}$ (by ~83.4%; 63.7%), and AOC (by ~46.7%; 27.7%). This study elucidates the significant role of HONO in the atmospheric oxidation chemistry in the upper PBL and lower FT of the NCP region, where previous studies mainly focused on HONO at the ground level.

## Acknowledgements

The authors thank all the staff of the Taishan National Reference Climatological Station for their logistics and help during the field observations. We are grateful for the MCM group of the Leeds University for provision of the MCM model. This work was sponsored by the National Key Research and Development Program of China (2016YFC0200500), National Natural Science Foundation of China (41922051 and 91544213), Shandong Provincial Science Foundation for Distinguished Young Scholars (ZR2019JQ09), the Jiangsu Collaborative Innovation Center for Climate Change, and the Taishan Scholars (ts201712003).

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


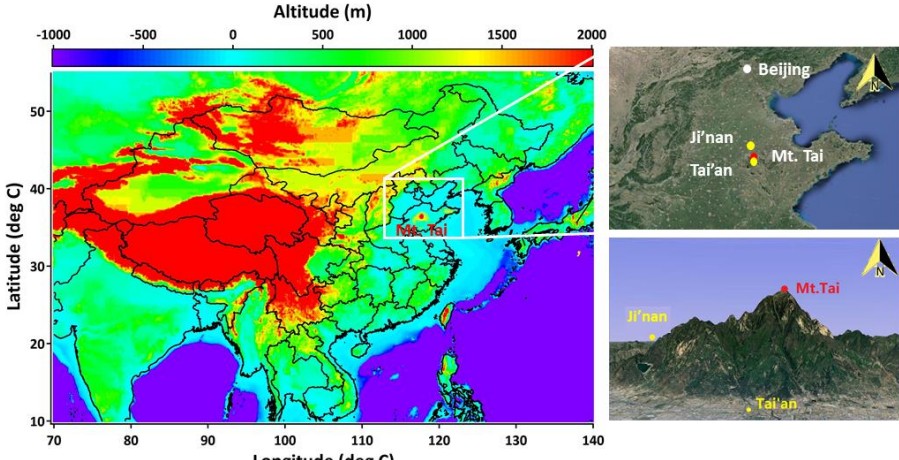

**Figure 1.** Maps showing the locations of Mt. Tai and the nearby cities. The left map is color-coded by the geographical height from the ETOPO1 Global Relief Model (https://www.ngdc.noaa.gov/mgg/global/). The right-top map represents the North China Plain and the right-bottom map is the 3-D shape of Mt. Tai with nearby urban regions.

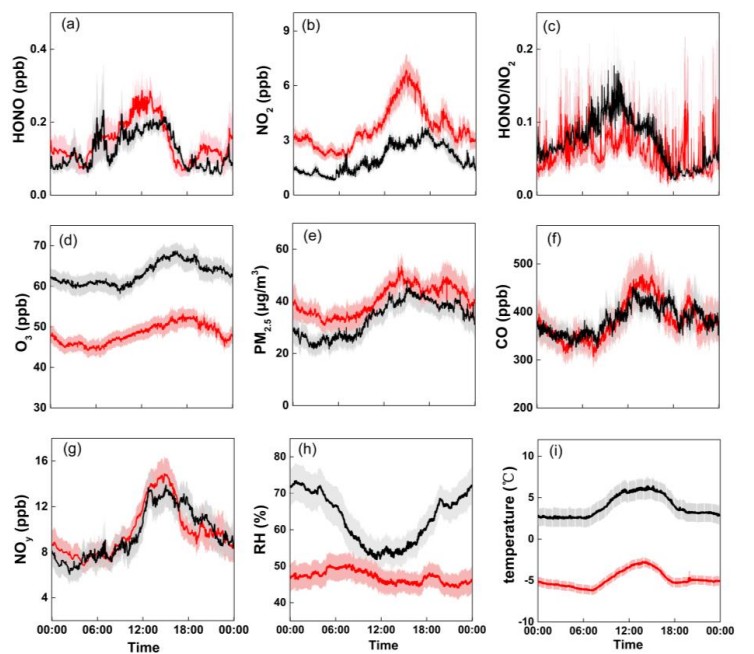

**Figure 2.** Average diurnal variations of (a) HONO, (b) $NO_2$, (c) $HONO/NO_2$, (d) $O_3$, (e) $PM_{2.5}$, (f) CO, (g) $NO_y$, (h) RH, and (i) temperature in winter (red) and spring (black) at Mt. Tai. Shaded area indicates the standard error of measurement data.

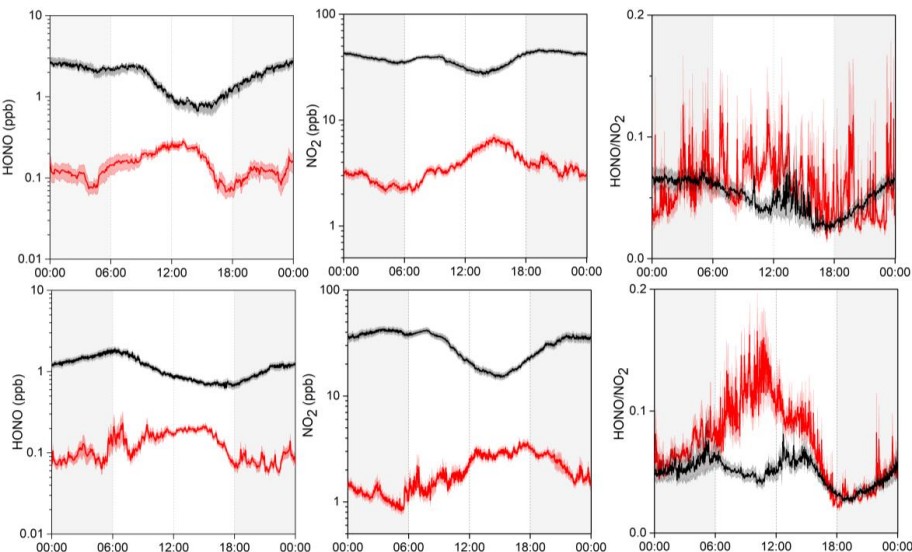

**Figure 3.** Comparison of the average diurnal variations of HONO, NO$_2$, HONO/NO$_2$ at Mt. Tai (red) and in urban Ji'nan (black) in winter (upper panel) and spring (lower panel). Shaded area indicates the standard error of measurement data.

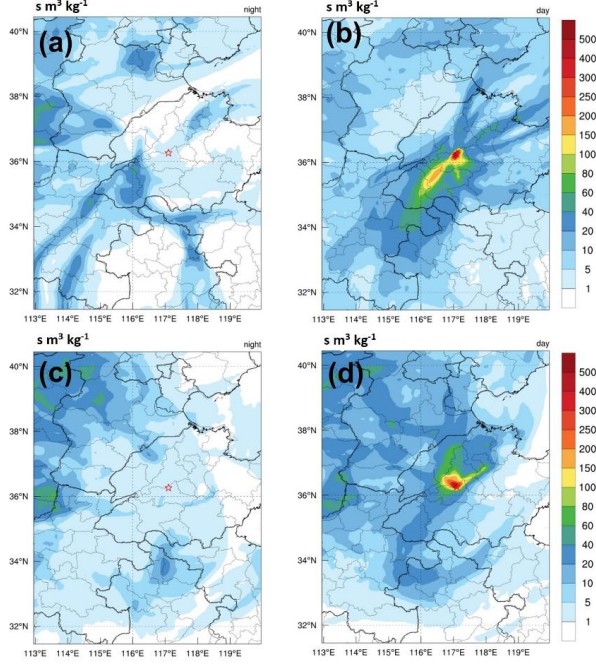

**Figure 4.** Average distribution of the "footprint" retroplume of the 3-day backward trajectory during (a) spring nighttime (2:00-08:00 LT) and (b) spring afternoon (12:00-18:00 LT); (c) winter nighttime (2:00-8:00 LT) and (d) winter afternoon (12:00-18:00 LT). Note that the horizontal resolution is 0.1°.

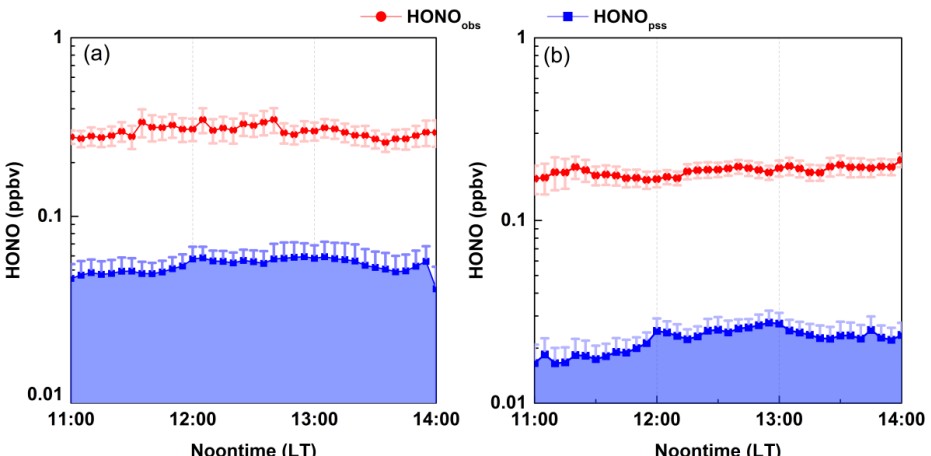

**Figure 5.** Average noontime [HONO]$_{pss}$ concentrations compared with the observed HONO concentrations in (a) winter and (b) spring. Shaded area indicated the level of [HONO]$_{pss.}$ Error bar indicated the standard error of data.

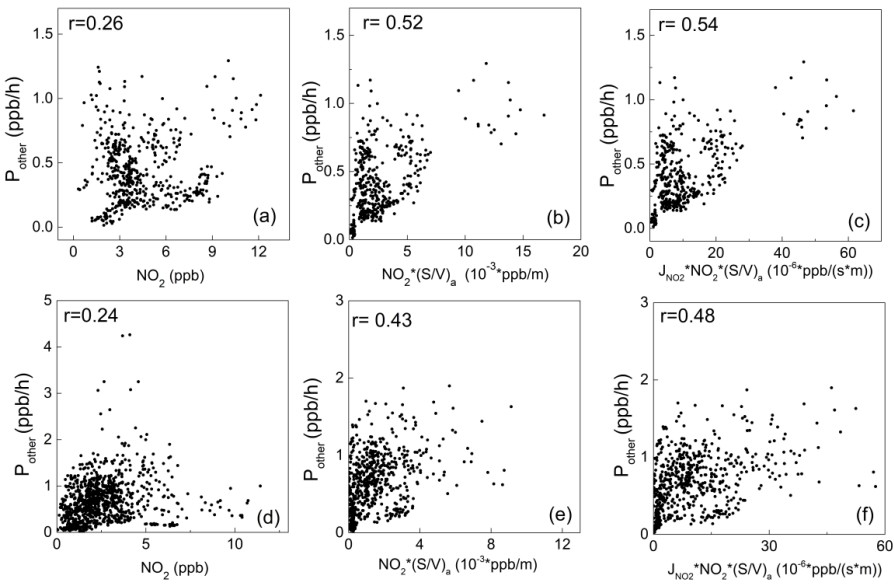

**Figure 6.** Scatter plots of the additional daytime HONO source strength (P$_{other}$) with NO$_2$, NO$_2$*(S/V)$_a$, and J$_{NO2}$*NO$_2$* (S/V)$_a$ in winter (upper panel) and spring (lower panel).





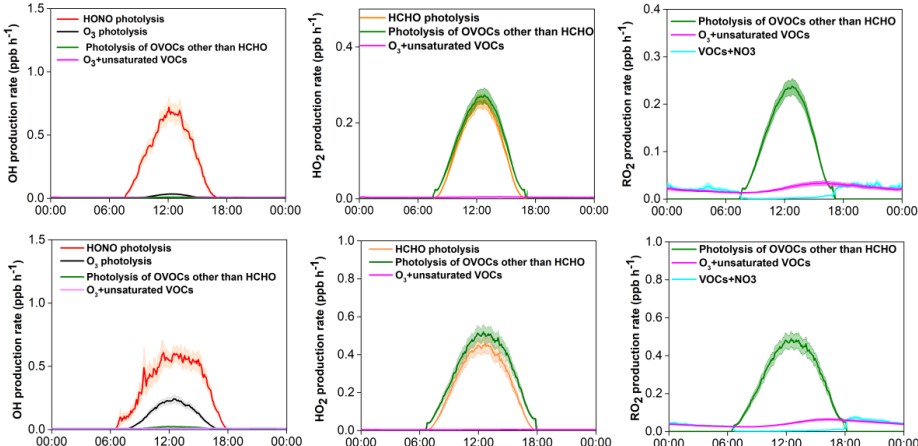

**Figure 7**. Model-simulated average diurnal variations of major primary sources of OH, HO$_2$ and RO$_2$ at Mt. Tai in winter (upper panel) and spring (lower panel). Shaded area indicates the standard error of the data.

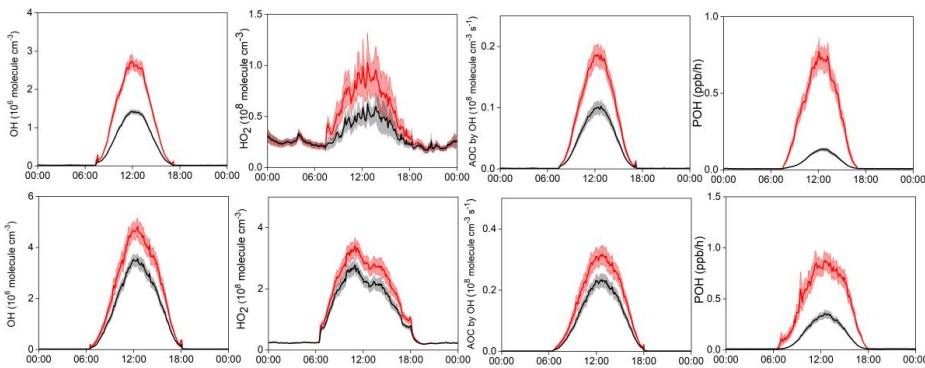

**Figure 8.** Model-simulated average diurnal variations of OH, HO$_2$, AOC, and primary OH production rate (P$_{OH}$) at Mt. Tai in winter (upper panel) and spring (lower panel), with (red) and without (black) constraints of HONO measurement data. Note that the model only considers the homogenous HONO formation from OH+NO reactions. Shaded area indicates the standard error.





**Table 1.** Statistics (average ± standard deviation) of trace gases, PM$_{2.5}$, and meteorological parameters measured at Mt. Tai in winter 2017 and spring 2018.

| Parameter | 2017 winter | | | 2018 spring | | |
|---|---|---|---|---|---|---|
| | All data | Daytime | Nighttime | All data | Daytime | Nighttime |
| HONO (ppb) | 0.15±0.15 | 0.18±0.15 | 0.11±0.14 | 0.13±0.15 | 0.16±0.18 | 0.09±0.12 |
| NO$_2$ (ppb) | 3.70±3.24 | 4.26±3.59 | 3.10±2.71 | 2.00±1.91 | 2.29±1.99 | 1.72±1.77 |
| NO (ppb) | 0.77±1.44 | 1.27±1.71 | 0.05±0.05 | 0.26±0.35 | 0.39±0.40 | 0.09±0.12 |
| NO$_X$ (ppb) | 4.70±4.40 | 5.68±5.08 | 3.26±2.58 | 2.18±2.09 | 2.58±2.26 | 1.77±1.81 |
| NO$_y$ (ppb) | 9.83±7.35 | 11.03±7.28 | 8.61±7.22 | 9.61±7.31 | 10.61±7.62 | 8.63±6.84 |
| O$_3$ (ppb) | 48±12 | 49±12 | 48±12 | 63±14 | 64±14 | 63±14 |
| PM$_{2.5}$ (µg/m$^3$) | 40.3±27.5 | 41.2±24.8 | 39.5±29.8 | 33.7±26.7 | 35.9±26.3 | 31.6±27.0 |
| SO$_2$ (ppb) | 3.7±3.9 | 3.7±2.0 | 3.7±5.1 | 1.6±1.6 | 2.1±1.8 | 1.1±1.2 |
| CO (ppb) | 382±210 | 398±227 | 366±192 | 382±161 | 395±162 | 170±159 |
| RH (%) | 46±20 | 46±20 | 46±21 | 60±26 | 55±24 | 66±27 |
| Temperature (℃) | -4.6±3.7 | -4.0±3.8 | -5.2±3.5 | 4.3±6.4 | 5.1±6.5 | 3.2±6.0 |
| Pressure (hpa) | 849.0±0.3 | 848.9±0.3 | 849.1±0.3 | 846.9±0.3 | 847.1±0.3 | 846.7±0.3 |

Daytime period: 06:00-18:00 LT; Nighttime period: 18:00-06:00 LT.



**Table 2.** Comparison of ambient HONO concentration levels observed at Mt. Tai with previous studies at both surface and high-elevation sites.

| Type | Site Location | Period | HONO (ppb) Mean±SD | References |
|---|---|---|---|---|
| | Beijing, China (urban) | Sep -Oct 2015 | 2.27±1.82 | Wang et al., 2017 |
| | Beijing, China (urban) | 3 Jan -27 Jan 2016 | 1.05±0.89 | Wang et al., 2017 |
| | Beijing, China (urban) | Apr.-May 2016 | 1.05±0.95 | Wang et al., 2017 |
| | Beijing, China (urban) | Jun-July 2016 | 1.38±0.90 | Wang et al., 2017 |
| | Ji'nan, China (urban) | Sep-Nov 2015 | 0.78±0.60 | Li et al., 2018a |
| | Ji'nan, China (urban) | Dec 2015-Feb 2016 | 1.75±1.62 | Li et al., 2018a |
| Ground level | Ji'nan, China (urban) | Mar-May 2016 | 1.16±0.90 | Li et al., 2018a |
| observations in China | Ji'nan, China (urban) | Jun-Aug 2016 | 1.12±0.93 | Li et al., 2018a |
| | Wangdu, China (rural) | Jun-Jul 2014 | 0.91±0.48 | Liu et al., 2019 |
| | Guangzhou, China (urban) | Jun 2006 | 2.80 | Qin et al., 2009 |
| | Xinken, China (rural) | 13 Oct-2 Nov.2004 | 1.20 | Su et al., 2008b |
| | Back Garden, China (rural) | Jul 2006 | 0.76 | Li et al., 2012 |
| | Shanghai, China (urban) | Oct 2004-Jan 2005 | 1.1±1.0 | Cui et al., 2018 |
| | Whiteface Mountain, USA (1483 m a.s.l.) | 14 Jun-20 Jul 1999 | 0.046 | Zhou et al., 2007 |
| | Summit, Greenland (3200 m a.s.l.) | 2 Jul-4 Jul 1999 | 0.009 | Dibb et al., 2002 |
| | Hohenpeissenberg, Germany (980 m a.s.l.) | 03 Jul – 12 Jul 2002 | 0.039 | Acker et al., 2006 |
| | Mt.Brocken, Germany (1142 m a.s.l.) | 29 Jun –14 Jul 2004 | 0.063 | Acker et al., 2006 |
| High-altitude | North Atlantic Ocean (<2500 m a.g.l.) | 19 Jun –4 Jul 1999 | 0.056 | Acker et al., 2001 |
| Observations around the world | Northern Michigan (1000 m-1900 m.) | 5 July 2013 | 0.011±0.002 | Ye et al., 2016 |
| | Northern Italy (300-1000 m a.g.l.) | 8 July 2013 | 0.009±0.002 | Ye et al., 2016 |
| | Southeastern US (>1500 m a.g.l.) | 30 Jul - 6 Aug 2007 | 0.009 | Zhang et al., 2009 |
| | Mt. Tai, China (1534 m a.s.l) | 12 Jul 2012 | ~0.15 | Li et al., 2014 |
| | | 1 Jun –15 Jul 2013 | 0.006±0.003 | Ye et al., 2018 |
| | | Nov-Dec 2017 | 0.15±0.15 | This study |
| | | Mar-Apr 2018 | 0.13±0.15 | This study |



**Table 3**. Summary of major primary sources of OH, $HO_2$ and $RO_2$ radicals simulated by MCM model.

| Radical primary source [a] | 2017 winter | | 2018 spring | |
|---|---|---|---|---|
| | Rate [b] (ppb/h) | Proportion (%) [c] | Rate (ppb/h) [b] | Proportion (%) [c] |
| **$P_{OH}$** | | | | |
| O₃ photolysis | 0.022 | 1.9 | 0.178 | 9.0 |
| HONO photolysis | 0.520 | 44.4 | 0.511 | 25.8 |
| OVOCs photolysis | 0.007 | 0.6 | 0.018 | 0.9 |
| O₃ + unsaturated VOCs | 0.009 | 0.7 | 0.010 | 0.5 |
| **$P_{HO2}$** | | | | |
| HCHO photolysis | 0.193 | 16.5 | 0.374 | 18.9 |
| OVOCs photolysis | 0.210 | 17.9 | 0.434 | 21.9 |
| O₃ + unsaturated VOCs | 0.004 | 0.4 | 0.005 | 0.2 |
| **$P_{RO2}$** | | | | |
| OVOCs | 0.182 | 15.5 | 0.407 | 20.5 |
| VOCs+NO₃ | 0.002 | 0.2 | 0.004 | 0.2 |
| O₃ + unsaturated VOCs | 0.023 | 2.0 | 0.042 | 2.1 |

[a] OVOCs photolysis denotes the photolysis of OVOCs other than formaldehyde.

[b] Rate is the mid-day (9:00-15:00 LT) average radical production rate.

[c] Proportion is the ratio of the target radical source to the total quantified radical primary production rates.