# Peer review of "Sources of nitrous acid (HONO) in the upper boundary layer and lower free troposphere of the North China Plain: insights from the Mount Tai Observatory"

_Atmospheric Chemistry and Physics, 2020_

## Referee Comment (RC1) · Anonymous Referee #1 · 16 Jun 2020

The paper reports measurements of nitrous acid HONO at Mt Tai, a mountain site above the North China Plane in the Winter and Summer. The measurements were compared to the output of an MCM chemical box model to look for clues as to the processes that produce HONO values above those predicted by the NO-OH photostationary state (PSS). The measurements are interesting and their comparison to HONO measurements made at nearby surface sites are useful. I would like to see the authors expand their thinking about possible interferences in the chemical measurement, and use the MCM model to examine those. It is unfair to expect the authors to resolve these issues in this context, so I think the paper should be acceptable after some of that material is added and after the resolution of the following general and specific comments.

I have only a few specific comments/changes, and then I'd like you to consider some general issues related to using LOPAP for measuring HONO, especially in Winter.

General Comments: Wintertime HONO measurements must consider the possibility of peroxynitric acid, HO2NO2, interferences. HO2NO2 is soluble in aqueous solution and forms HONO/NO2- readily on surfaces and in aqueous solution. In addition, HO2NO2 is going to be favored at low temperatures when there is substantial HOx/NOx photochemistry (Veres et al., 2015), which is the situation at Mt Tai in the Wintertime and Springtime. There is one study that has shown the HO2NO2 interference in the LOPAP method to be about 15% (Legrand et al., 2014). However, there is at least one other data set that implies the interference could be higher than that, see the Supplemental Material of Rappenglück, et al, (2014) which describes Wintertime LOPAP measurements in the middle of an oil and gas field when intense O3 photochemical production was happening. One important piece of information in this regard is that modeling of this Wintertime photochemistry found that O3 was overpredicted by substantial amounts when the LOPAP measured HONO was used in the model compared to PSS HONO (Carter and Seinfeld, 2012).

I am not expecting the authors to resolve this issue in this work. However, since this paper has extensive MCM modeling, I would like the authors to explore several questions: What are the HO2NO2 levels predicted by their model and how do they compare to the "excess HONO? What is the O3 predicted when using both the PSS HONO and the LOPAP measured HONO and how do those levels compare to measured O3?

Modeled ozone levels that are much higher than measured would be clue that the HONO measurement has an interference. I would also note that the HO2NO2 source is HO2 + NO2, so one would expect the extra HONO (above PPS HONO) to scale with NO2 and JNO2. How does the quantity [NO2]*JNO2 correlate with PHONO?

Specific Comments:

The data used in this paper, and ideally the code used for the model, must be made

available to the community. Please deposit your data in an acceptable repository (see the ACP Instructions to Authors), or an accessible repository of your choosing. If the model code is already generally accessible, please specify where it may be obtained.

Technical Comments/Corrections

Page 1, Line 16: should be "conducted at the surface" Page 1, Line 20: Are these averages and standard deviations? Page 1, Line 21: Should be "with broad noontime maxima" Page 1, Line 29: the statement about HOx radical levels is misleading. You don't have actual measurements of HOx radicals, only two different model cases, one based on PSS HONO and the other base on what the model says HOx would be given LOPAP measured HONO. You need to be clear about how you talk about it. When you say "underestimated" you are implying that the higher modeled HOx is in some sense "true" or "correct", when really, it's only a different estimate. This language is found other places in the paper and needs to be changed. Page 2, Line 9: Should be "affects human health" Page 2, Line 19: The PROPHET site is 238m in elevation, so is not a high altitude site. Page 5, Line 12: What does "SHARP" stand for? Page 6, Line 1: What are the uncertainties in the JNO2 measurements and estimates? Page 6, Line 5: What does FNL stand for? Page 6, Line 27: The definition of AOC is hard to follow based on this description. At first I thought the authors meant Sum{kOH[Xi]}, where [Xi] is the concentration of the individual species listed. That is properly termed "OH reactivity". I think the authors mean Sum{kOH[OH][Xi]}, but they need to make that explicitly clear. Page 7, Line 22: I don't understand what the authors mean by "inspection of data reveals the higher than expected concentration levels of HONO". At this point in the paper, we have no context with which to judge this, i.e. we don't know what PSS HONO is or what [HONO] at remote sites might be expected. Page 8. Line 21: It seems to me, the authors could use a tracer to more precisely determine the timing of upslope arrival at the site. Page 10, Line 1: I think there are better references for this than Donahue et al., (1973). Page 10, Line 26. The phrase the "air masses …... facilitate a pseudo-steady state" doesn't make sense. The short

lifetimes facilitate steady state. Page 12, Line 16: "dissected" seems like the wrong word here. I think "examined" or "explored" would be better. Page 13, Lines 20, 21: You are phrasing these results like you know what HOx and AOC is or should be, when really, you are comparing two different model results, one with and one without the additional HONO implied by the LOPAP measurements. The same kind of language is used in the Conclusions. These statements should be rephrased. Page 13, Baergen and Donaldson reference – this is a talk not a journal publication. I am fairly sure this group has a journal publication about this.

References

Carter, W. P. L. and Seinfeld, J. H.: Winter ozone formation and VOC incremental reactivities in the Upper Green River Basin of Wyoming Atmos. Environ., 50, 255-266, 2012.

Legrand, M., Preunkert, S., Frey, M., Bartels-Rausch, T., Kukui, A., King, M. D., Savarino, J., Kerbrat, M., and Jourdain, B.: Large mixing ratios of atmospheric nitrous acid (HONO) at Concordia (East Antarctic Plateau) in summer: a strong source from surface snow?, Atmos. Chem. Phys., 14, 9963-9976, 2014.

Rappenglück, B., Ackermann, L., Alvarez, S., Golovko, J., Buhr, M., Field, R., Soltis, J., Montague, D. C., Hauze, B., Adamson, S., Risch, D., Wilkerson, G., Bush, D., Stoeckenius, T., and Keslar, C.: Strong wintertime ozone events in the Upper Green River Basin, Wyoming, Atmos. Chem. Phys., 14, 4909-4934, 2014.

Veres, P. R., J.M., R., Wild, R., Edwards, P. M., Brown, S. S., Bates, T. S., Quinn, P. K., Johnson, J. E., Zamora, R. J., and de Gouw, J. A.: Peroxynitric acid (HO2NO2) measurements during the UBWOS 2013 and 2014 studies using iodide ion chemical ionization mass spectrometry, Atmos. Chem. Phys., 15, 8101-8114., 2015.

---

## Referee Comment (RC2) · Anonymous Referee #2 · 10 Jul 2020

This manuscript describes measurements of Nitrous Acid (HONO) and a suite of other trace gases at a mountaintop site (Mt. Tam Observatory) on the North China Plain. Their goal was to assess whether currently understood HONO sources could explain measured HONO mixing ratios at the (sometimes), free tropospheric site. The authors use their observations coupled with model output to calculate midday HONO photo-stationary state values (PSS) and compare their observations with Master Chemical Mechanism (MCM) model output. Closing the HONO budget has been a challenging long-standing issue within the community and this paper is a further attempt to do so. The publication is generally well written and the content falls within the scope of Atmospheric Chemistry and Physics. I recommend publication once the following comments

are addressed.

Major Comments:

While the Long Path Absorption Photometer Technique (LOPAP) has been extensively tested for a variety of interferences (Heland et al., 2001;Kleffmann and Wiesen, 2008), it has been shown to have an interference from peroxynitric acid (HO2NO2, PNA), (Legrand et al., 2014). While in many studies ambient temperatures would have rendered this potential interference negligible, I cannot help but feel this is the type of scenario where it could potentially be a problem. The temperatures under which these measurements were performed (and NOx levels) should support a long enough PNA lifetime for it to be observed. In addition, the authors indicate that the missing HONO source should scale with solar radiation and NO2. PNA should also scale with both of these parameters. While it would be nice to have a full interference test of the instrument for PNA, synthesizing or developing photolytic generation sources can be a challenge. The authors should address this by adding a section discussing the possibility of the interference. Using their MCM model runs they could provide an estimate of what best/worst case scenarios would look like. I doubt that PNA alone is the missing "HONO" source, however constraining the possibility of interference could give a more accurate picture to how much HONO is in fact missing.

Specific Comments:

Please indicate where the data and model output are available to readers. Title: Should be "The North China Plain"? P1 L20: Are the mixing ratios means or medians? What are they? I believe it is mentioned further in the text, but it should be noted here as well. P1 L21: Could the noontime max be biased by measuring HO2NO2 (see major comments)? P1 L27: Would a source consistent with NO2 not also be consistent with HO2NO2 (see major comments)? P4 L9: Change the wording of "Observatory has been widely deployed". You don't really deploy a building. Perhaps "Observatory has been widely used as a sampling location" ? P6 L22: What do the authors mean by

approximated by the CO (Temperature) data? Was it approximated by CO or temperature? Please clarify. P7 L27: I find the description of AOC to be confusing. Are the authors simply trying to describe OH-Reactivity or something else? Please clarify. P7 L8: This wording is a bit ackward "The atmospheric conditions at Mt. Tai were featured by a cold and dry weather". Perhaps "The atmospheric conditions at Mt. Tai were dominated by cold and dry weather."? P9 L18: "It can be argued that the heterogeneous formation of HONO should be stronger at the mountaintop, which may be due to the more intense solar radiation at the high altitudes." Why can this be argued? The ratios are almost within one standard deviation of each other. This would potentially be true assuming the source scales with light intensity, but what would the difference in surface area for the heterogeneous reaction be? That's not clear.

P9 L31: Is there no newer reference than 1973? Surely there have been some advances.

P10 L6: The surface HONO would most certainly be extensively diluted by the time it reached the elevated site. Could the authors not use a conserved tracer to estimate the dilution? Or for that matter the upslope time?

P13 L20: How do you know the model underestimated the HOx radical levels? I am unaware of HOx measurements as part of this campaign constraining this. They aren't noted anywhere. This (as well as in the conclusion) should be re-worded.

Figure 2: While the shaded areas currently represent standard error of the measurement, perhaps showing standard deviation of the measurement might be more useful so that the reader can get a better feel for the range of values observed.

Figure 5: A figure showing the relative contribution of each of known HONO formation pathways would also be nice so that readers can get a visual idea of the importance of each pathway at this location.

References:

Heland, J., Kleffmann, J., Kurtenbach, R., and Wiesen, P.: A New Instrument To Measure Gaseous Nitrous Acid (HONO) in the Atmosphere, Environmental Science & Technology, 35, 3207-3212, 2001.

Kleffmann, J., and Wiesen, P.: Technical Note: Quantification of interferences of wet chemical HONO LOPAP measurements under simulated polar conditions, Atmos.Chem.Phys., 8, 6813-6822, 2008.

Legrand, M., Preunkert, S., Frey, M., Bartels-Rausch, T., Kukui, A., King, M. D., Savarino, J., Kerbrat, M., and Jourdain, B.: Large mixing ratios of atmospheric nitrous acid (HONO) at Concordia (East Antarctic Plateau) in summer: a strong source from surface snow?, Atmos. Chem. Phys., 14, 9963-9976, 10.5194/acp-14-9963-2014, 2014.

---

## Referee Comment (RC3) · Anonymous Referee #3 · 10 Jul 2020

The work by Jiang et al presents measurements of HONO and other supporting species at Mt Tai, a mountaintop site in the North China Plain. Concurrent ground level measurements were also performed, and comparison were made to explore the source differences. The authors present an interesting data set and provide modelling work to aid their interpretation. Overall, this paper is well thought out and written, with the results clearly presented in the tables and figures. I would recommend publication after consideration to the comments below.

Minor comments Section 2.2: Were the same instruments used at Mt Tai as the ground-level monitoring stations? This is particularly important for HONO measurements, as

previous work has shown significant differences can be reported for co-located HONO instruments, even of the same type (See e.g. Crilley et al., 2019). Furthermore, were there any inter-comparison measurements of the instruments at Mt Tai and ground to account for any differences between instruments that may affect the comparison? Page 5, line 3: Typically, the baseline is measured every 4-8 hours with a LOPAP (see e.g. Crilley et al., 2019; Kleffmann and Weisen 2008) to capture the temporal variability. Measuring the baseline every 11h 30min may not be sufficient to capture the baseline variability and I am curious why the authors chose to do it like this. Page 6, line 22: I don't quite follow this sentence 'based on the CO (temperature) data and the measured correlations with CO (temperature) for anthropogenic (biogenic) VOCs' What do you mean by CO (temperature)? Page 10, line 23: are the reported j(HONO) and OH concentrations noontime maxima or daily averages? Page 12, line 7: Here you state that heterogenous reaction with NO2 on aerosol surfaces should be a significant daytime HONO source at Mt. Tai rather than on ground. What is ratio of ground vs aerosol surface area? I am asking to try and understand why this NO2 reaction may preferentially occur on aerosol surfaces at Mt Tai, unlike previous work at ground level. Page 13, line 10: it would also be good to report the percentage HONO photolysis contributes to OH production from the model, as this would enable comparison to other work. Figure 3: Why is there so much more noise in the HONO/NO2 diurnal plots compared to HONO and NO2? Figure 5: I am surprised that there is no noon-time maxima in HONO pss, as the OH should peak then (as seen in Fig 8) along with the NOx? (as shown in Fig2)?

References Crilley, L. R., Kramer, L. J., Ouyang, B., Duan, J., Zhang, W., Tong, S., Ge, M., Tang, K., Qin, M., Xie, P., Shaw, M. D., Lewis, A. C., Mehra, A., Bannan, T. J., Worrall, S. D., Priestley, M., Bacak, A., Coe, H., Allan, J., Percival, C. J., Popoola, O. A. M., Jones, R. L., and Bloss, W. J.: Intercomparison of nitrous acid (HONO) measurement techniques in a megacity (Beijing), Atmos. Meas. Tech., 12, 6449–6463, https://doi.org/10.5194/amt-12-6449-2019, 2019. Kleffmann, J. and Wiesen, P., 2008. Quantification of interferences of wet chemical HONO LOPAP measurements

under simulated polar conditions. Atmospheric Chemistry & Physics, 8(22).

---

## Author Comment (AC1) · 18 Aug 2020

**Response to Reviewer 1:**

*The paper reports measurements of nitrous acid HONO at Mt Tai, a mountain site above the North China Plane in the Winter and Summer. The measurements were compared to the output of an MCM chemical box model to look for clues as to the processes that produce HONO values above those predicted by the NO-OH photostationary state (PSS). The measurements are interesting and their comparison to HONO measurements made at nearby surface sites are useful. I would like to see the authors expand their thinking about possible interferences in the chemical measurement, and use the MCM model to examine those. It is unfair to expect the authors to resolve these issues in this context, so I think the paper should be acceptable after some of that material is added and after the resolution of the following general and specific comments.*

Response: We thank the reviewer for the helpful comments to improve our original manuscript. We have carefully considered all the review comments and revised the manuscript accordingly. Especially, we adopted the constructive suggestion to examine the possible interference from $HO_2NO_2$ by the MCM modeling analysis. For clarity, we list the original reviewer's comments below *in black italic*, and provide our responses and changes in the manuscript in blue and red, respectively.

***General comments***

*Wintertime HONO measurements must consider the possibility of peroxynitric acid, $HO_2NO_2$, interferences. $HO_2NO_2$ is soluble in aqueous solution and forms $HONO/NO_2^-$ readily on surfaces and in aqueous solution. In addition, $HO_2NO_2$ is going to be favored at low temperatures when there is substantial $HO_x/NO_x$ photochemistry (Veres et al., 2015), which is the situation at Mt Tai in the Wintertime and Springtime. There is one study that has shown the $HO_2NO_2$ interference in the LOPAP method to be about 15% (Legrand et al., 2014). However, there is at least one other data set that implies the interference could be higher than that, see the Supplemental Material of Rappenglück, et al, (2014) which describes Wintertime LOPAP measurements in the middle of an oil and gas field when intense $O_3$ photochemical production was happening. One important piece of information in this regard is that modeling of this Wintertime photochemistry found that $O_3$ was overpredicted by substantial amounts when the LOPAP measured HONO was used in the model compared to PSS HONO (Carter and Seinfeld, 2012).*

*I am not expecting the authors to resolve this issue in this work. However, since this paper has extensive MCM modeling, I would like the authors to explore several questions: What are the $HO_2NO_2$ levels predicted by their model and how do they compare to the "excess HONO? What is the $O_3$ predicted when using both the PSS HONO and the LOPAP measured HONO and how*

*do those levels compare to measured $O_3$?*

*Modeled ozone levels that are much higher than measured would be clue that the HONO measurement has an interference. I would also note that the $HO_2NO_2$ source is $HO_2 + NO_2$, so one would expect the extra HONO (above PPS HONO) to scale with $NO_2$ and $J(NO_2)$. How does the quantity $[NO_2]*J(NO_2)$ correlate with $P_{HONO}$?*

**Response:** We thank the reviewer for the constructive comment that we did not consider carefully before. According to the suggestions, we reviewed more literatures and performed more MCM simulations to carefully explore the potential interferences from $HO_2NO_2$ on our HONO measurements. Below are some detailed results and our thoughts about this issue.

(1) We simulated the time series of $HO_2NO_2$ at Mt. Tai in winter and spring by the MCM chemical box model constrained with all our measured species including HONO. As shown from Figure R1-1, moderate concentration levels of $HO_2NO_2$ were predicted by the model at Mt. Tai, with average values (±SD) of 0.07±0.06 ppbv and 0.03±0.04 ppbv in winter and spring, respectively. If we took the $HO_2NO_2$ interference of 15% reported by Legrand et al. (2014), the potential interference to the excess HONO (measured HONO minus the PSS HONO) were 16%±15% and 11%±10% in winter and spring, respectively. If we assumed a 100% of interference (representing the worst case), the potential interference from $HO_2NO_2$ to the excess HONO were 72%±30% and 66%±54% in winter and spring.

[Figure]

**Figure R1-1.** Model simulated $HO_2NO_2$ concentrations (grey) and comparison with the measured HONO (red) and excess HONO (green) at Mt. Tai in winter and spring.

(2) The observed $O_3$ concentrations at such a mountain site is mainly dominated by transport due to the mountain-valley breeze and strong winds at the mountaintop. It is really difficult for

a chemical box model with little consideration of physical processes to reproduce the observed variation pattern of $O_3$ at Mt. Tai. Although the model cannot reproduce the nighttime $O_3$ level observed at Mt. Tai, the modelled afternoon $O_3$ maxima (with measured HONO as constraints) were comparable to the observed $O_3$ peak levels (see Fig. R1-2 for examples with relatively weak winds). It should be noted that the modelling analyses presented in this study have used measured $O_3$ data (and other radical precursors) as constraints to estimate their impacts on the radical production and atmospheric oxidation capacity.

[Figure]

**Figure R1-2.** Model-simulated $O_3$ with HONO constraint (black) and without HONO constraint (HONO PSS; red) and comparison with the measured data (blue) on two cases with relatively weak daytime winds.

(3) We examined the relationship between excess HONO (the measured HONO minus the PSS HONO) and $[NO_2]*J(NO_2)$, and the results are shown in Fig. R1-3. As we can see, the correlation was overall rather weak between excess HONO and $[NO_2]*J(NO_2)$, especially in winter. In comparison, the correlations were improved after the aerosol surface area was taken into consideration, with r of 0.54 and 0.48 between excess HONO and $[NO_2]*J(NO_2)*(S/V)a$ in winter and spring (see Fig. 6 in the manuscript). This indicates that the interference from $HO_2NO_2$ may not be a major factor in the determined excess HONO, and aerosol surface should play an important role.

[Figure]

**Figure R1-3.** Scatter plots of excess HONO versus $[NO_2]*J(NO_2)$ at Mt. Tai in (a) winter and (b) spring.

(4) Based on the literature review, $HO_2NO_2$ is indeed a considerable interference to the LOPAP HONO measurements in the low temperature environments, such as polar regions and high mountain areas. Nonetheless, as the reviewer mentioned, current studies have not reached a consensus on the magnitude of $HO_2NO_2$ interference to the LOPAP measurements. For example, Legrand et al. (2014) reported that the $HO_2NO_2$ interference measured by LOPAP was about 15% in their lab experiments. Rappenglück et al. (2014) suggested that the interference could be higher than 15% from their wintertime measurements in an oil and gas field when intense $O_3$ production happened. Kerbrat et al. (2012) reported that an unpublished work by Ammann showed the $HO_2NO_2$ interference to the LOPAP was less than 3%. Obviously, more experiments are still needed to quantify the potential interference from $HO_2NO_2$ to the LOPAP measurements.

In the revised manuscript, we have added a section (see below) to discuss the possible interference from $HO_2NO_2$ to the measured HONO in the present study.

**"Possibility of measurement interference from peroxynitric acid (PNA; $HO_2NO_2$):** While the LOPAP instrument has been extensively tested for a variety of interferences (Heland et al., 2001; Kleffmann and Wiesen, 2008), some recent studies reported that it may be subject to positive interference from $HO_2NO_2$ (e.g., Legrand et al., 2014). Due to the thermo decomposition nature of $HO_2NO_2$, its interference is generally negligible at ambient temperatures at the ground level, but may become important in the circumstances with low temperature and active photochemistry. Legrand et al. (2014) reported that the interference from $HO_2NO_2$ to their HONO measurements was about 15% according to laboratory experiments. In the present study, we did not conduct in-situ measurements of $HO_2NO_2$. To estimate the potential interference for our HONO measurements, we simulated the $HO_2NO_2$ concentrations at Mt. Tai in both campaigns by the MCM chemical box model constrained with all measured species including HONO. Figure S6 shows the time series of modelled $HO_2NO_2$ and its comparison with the measured HONO and missing HONO (measured HONO minus [HONO]$_{pss}$) concentrations. Overall, moderate concentration levels of $HO_2NO_2$ were predicted by the model at Mt. Tai, with average values (±SD) of 0.07±0.06 ppbv and 0.03±0.04 ppbv in winter and spring, respectively. If we took the $HO_2NO_2$ interference of 15% (Legrand et al., 2014), the potential interference to the missing HONO were 16%±15% and 11%±10% in winter and spring, respectively. Figure S7 shows the scatter plots of missing HONO versus [NO$_2$]*J(NO$_2$), an indicator of the $HO_2NO_2$ production. As we can see, the correlation was rather weak between missing HONO and [NO$_2$]*J(NO$_2$), especially in winter (r=0.19). This indicates that the interference from $HO_2NO_2$ may not be a major factor in the determined missing HONO, and more experiments are needed to confirm and quantify the possible interferences to the ambient HONO observations."

Kleffmann, J., and Wiesen, P.: Technical Note: Quantification of interferences of wet chemical

HONO LOPAP measurements under simulated polar conditions, Atmos. Chem. Phys., 8, 6813-6822, 2008.

Legrand, M., Preunkert, S., Frey, M., Bartels-Rausch, T., Kukui, A., King, M. D., Savarino, J., Kerbrat, M., and Jourdain, B.: Large mixing ratios of atmospheric nitrous acid (HONO) at Concordia (East Antarctic Plateau) in summer: a strong source from surface snow?, Atmos. Chem. Phys., 14, 9963-9976, 2014.

Rappenglück, B., Ackermann, L., Alvarez, S., Golovko, J., Buhr, M., Field, R. A., Soltis, J., Montague, D. C., Hauze, B., Adamson, S., Risch, D., Wilkerson, G., Bush, D., Stoeckenius, T., and Keslar, C.: Strong wintertime ozone events in the Upper Green River basin, Wyoming, Atmos. Chem. Phys., 14, 4909-4934, 2014.

Kerbrat, M., Legrand, M., Preunkert, S., Gallee, H., and Kleffmann, J.: Nitrous acid at Concordia (inland site) and Dumont d'Urville (coastal site), East Antarctica, J. Geophys. Res. Atmos., 117, 2012.

**Specific Comments**

*The data used in this paper, and ideally the code used for the model, must be made available to the community. Please deposit your data in an acceptable repository (see the ACP Instructions to Authors), or an accessible repository of your choosing. If the model code is already generally accessible, please specify where it may be obtained.*

**Response:** The measurement data used in the present study has been deposited in Mendeley Dataset (https://data.mendeley.com/datasets/wcn84cybx9/draft#folder-defadc56-944c-4f33-af54-14019d73ac61). The code used for the chemical box model was downloaded from the MCM website (http://mcm.leeds.ac.uk/MCMv3.3.1/home.htt), and was modified for the current location and period. The following statements have been added in the revised manuscript.

"Data availability. The measurement data and model output used in the present study can be accessed from https://data.mendeley.com/datasets/wcn84cybx9/draft#folder-defadc56-944c-4f33-af54-14019d73ac61. The code for the MCM model can be downloaded from the MCM website (http://mcm.leeds.ac.uk/MCMv3.3.1/home.htt)".

**Technical Comments/Corrections**

*1. Page 1, Line 16: should be "conducted at the surface"*

**Response:** Changed.

*2. Page 1, Line 20: Are these averages and standard deviations?*

**Response:** Yes. This statement has been revised as follows.

"HONO showed moderate concentration levels (average ± standard deviation: 0.15±0.15 and 0.13±0.15 ppbv), with maximum values of 1.14 and 3.23 ppbv in winter and spring, respectively."

*3. Page 1, Line 21: Should be "with broad noontime maxima"*

**Response:** Changed.

*4. Page 1, Line 29: the statement about $HO_x$ radical levels is misleading. You don't have actual measurements of $HO_x$ radicals, only two different model cases, one based on PSS HONO and the other base on what the model says $HO_x$ would be given LOPAP measured HONO. You need to be clear about how you talk about it. When you say "underestimated" you are implying that the higher modeled $HO_x$ is in some sense "true" or "correct", when really, it's only a different estimate. This language is found other places in the paper and needs to be changed.*

**Response:** We agree with the reviewer, and the original statements have been revised as follows in the revised version.

Page 1, Line 29: "The model only considering homogenous HONO source predicted much lower levels of the $HO_x$ radicals and atmospheric oxidation capacity, compared to the model constrained with measured HONO data."

Page 14, Line 26: "Clearly, the model only considering the homogeneous source and without observational constraints predicted much lower levels of the $HO_x$ radicals and AOC at Mt. Tai. Specifically, the discrepancy in the mid-day (9:00-15:00) average $P_{OH}$, OH, $HO_2$, and AOC can be up to 83.4% (63.7%), 47.2% (27.1%), 39.7% (20.3%), and 44.8% (24.9%) in winter (spring), compared to the base scenario with constraints of the measured HONO data."

Page 15, Line 19: "With only inclusion of the OH+NO reactions, significant reductions of the modelled OH (by ~47.2%; 27.1%), $HO_2$ (by ~39.7%; 20.3%), $P_{OH}$ (by ~83.4%; 63.7%), and AOC (by ~44.8%; 24.9 %) were found, compared with being constrained by observed HONO data."

*5. Page 2, Line 9: Should be "affects human health"*

**Response:** Changed.

*6. Page 2, Line 19: The PROPHET site is 238 m in elevation, so is not a high-altitude site.*

**Response:** Thanks for the suggestion. This reference has been deleted, and a new reference for a high-elevation site was added in the revised manuscript.

**"**Existing modelling studies may underestimate the AOC of high-altitude atmospheres owing to the lack of observational data constraints (Kukui et al., 2014).**"**

Kukui, A., Legrand, M., Preunkert, S., Frey, M. M., Loisil, R., Roca, J. G., Jourdain, B., King, M. D., France, J. L., and Ancellet, G.: Measurements of OH and $RO_2$ radicals at Dome C, East Antarctica, Atmos. Chem. Phys., 14, 12373-12392, 2014.

*7. Page 5, Line 12: What does "SHARP" stand for?*

**Response:** We have spell out "SHARP" in the revised manuscript as follows.

"The fine particle ($PM_{2.5}$) mass concentration was measured using a Synchronized Hybrid Ambient Real-time Particulate monitor (SHARP; *Thermo Scientific Model 5030*)."

*8. Page 6, Line 1: What are the uncertainties in the J(NO$_2$) measurements and estimates?*

**Response:** The $J(NO_2)$ monitor was mounted at the rooftop of the station and higher than all the other instruments' inlets (also without any shelter), and should be free from additional errors other than the monitor's inherent uncertainty. However, the in-situ $J(NO_2)$ observations were only available during the spring campaign. So, we had to estimate the $J(NO_2)$ for the winter campaign as well as J(HONO) and $J(O^1D)$ for both seasons, based on the concurrent $J(NO_2)$ observations and the TUV model calculations (scaling the TUV-calculated clear-sky J values with the ratio of measured $J(NO_2)$ to TUV $J(NO_2)$). Such estimation should be subject to some uncertainties, although this is the best what we can do with the available measurement data. The following statements have been added in the revised manuscript to elaborate the potential uncertainty of the estimation of J values.

"It should be noted that such estimation of J values is subject to some uncertainties, especially for those in winter when direct $J(NO_2)$ measurements were unavailable. Nonetheless, scaling the TUV-calculated clear-sky J values with the same ratio should not alter the major conclusion of this study regarding the impacts of HONO photolysis on the HOx sources and atmospheric

oxidation capacity."

*9. Page 6, Line 5: What does FNL stand for?*

**Response:** It is the final version of the NCEP reanalysis data and can be obtained from this website (https://rda.ucar.edu/datasets/ds083.3/index.html#sfol-fw?g=201608). The original statement has been revised as follows for clarity.

"The Weather Research and Forecasting (WRF) Model, driven by the NCEP FNL reanalysis data (https://rda.ucar.edu/datasets/ds083.3/index.html#sfol-fw?g=201608), was run to produce the high spatial resolution meteorological field."

*10. Page 6, Line 27: The definition of AOC is hard to follow based on this description. At first I thought the authors meant Sum{kOH[Xi]}, where [Xi] is the concentration of the individual species listed. That is properly termed "OH reactivity". I think the authors mean Sum{kOH[OH][Xi]}, but they need to make that explicitly clear.*

**Response:** We are sorry that the original description is misleading. Yes, it was defined as sum{$k_{OH}$[OH][Xi]}. For clarity, the original statements have been modified as follows in the revised manuscript.

"Also calculated by the model was the AOC by OH, which is defined here as the reaction rate of OH with NO, $NO_2$, $SO_2$, CO and VOCs (AOC = $\Sigma$($k_{OH}$[OH][Xi]): [Xi] is the concentration of the individual reactant species, and $K_{OH}$ is the rate coefficient of OH with Xi)."

*11. Page 7, Line 22: I don't understand what the authors mean by "inspection of data reveals the higher than expected concentration levels of HONO". At this point in the paper, we have no context with which to judge this, i.e. we don't know what PSS HONO is or what [HONO] at remote sites might be expected.*

**Response:** We agree with the reviewer, and this statement has been revised as follows in the revised version.

"The above inspection of data reveals the overall moderate HONO concentration levels as well as the frequent occurrence of HONO-laden plumes in the upper PBL and lower FT of the NCP region."

*12. Page 8. Line 21: It seems to me, the authors could use a tracer to more precisely determine the timing of upslope arrival at the site.*

**Response:** Thanks for the suggestion. As shown in Figure 2, most species including HONO, $NO_2$, NOy, $O_3$, CO and $PM_{2.5}$ showed daytime concentration peaks at Mt. Tai, confirming the upslope transport of boundary layer pollution to the mountaintop. To determine the timing of upslope arrival at the site, we chose CO as a tracer, which is relatively chemically inert and can represent the contribution of transport. During the two observation campaigns, the average CO concentrations increased from the morning and reached the maximum around noontime (e.g., 12:00-15:00 local time), which were almost coincided with the observed daytime HONO peak (~11:00-15:00 local time). The following statements have been added in the revised manuscript to elaborate this.

"The noontime HONO maximum (e.g., ~11:00-15:00 LT) at Mt. Tai suggested the potential upslope transport of HONO to the mountaintop and/or the presence of 'additional' daytime sources. The almost coincident noontime concentration peak of CO (e.g., ~12:00-15:00 LT) confirmed the upslope transport of boundary layer air to the mountaintop."

*13. Page 10, Line 1: I think there are better references for this than Donahue et al., (1973).*

**Response:** Thanks for the suggestion. A new and more recent reference has been cited in the revised version.

"Around noontime, the PBL has been developed and $K_z$ is generally in the order of $10^6$ $cm^2$ $s^{-1}$ (Zhang et al., 2009)."

"Zhang, N., Zhou, X., Shepson, P. B., Gao, H., Alaghmand, M., and Stirm, B.: Aircraft measurement of HONO vertical profiles over a forested region, Geophys. Res. Lett., 36, ,172-173, 2009."

*14. Page 10, Line 26. The phrase the "air masses ..... facilitate a pseudo-steady state" doesn't make sense. The short lifetimes facilitate steady state.*

**Response:** The original statement has been revised as follows in the revised version.

**"**Given such short lifetimes, the air masses arriving at Mt. Tai at noon should facilitate a steady state for HONO.**"**

*15. Page 12, Line 16: "dissected" seems like the wrong word here. I think "examined" or "explored" would be better.*

**Response:** It has been changed to "explored" in the revised version.

**"**The detailed chemical budget of $RO_x$ radicals was explored by the observation-based MCM box model.**"**

*16. Page 13, Lines 20, 21: You are phrasing these results like you know what HOx and AOC is or should be, when really, you are comparing two different model results, one with and one without the additional HONO implied by the LOPAP measurements. The same kind of language is used in the Conclusions. These statements should be rephrased.*

**Response:** We agree with the reviewer. These statements have been revised as follows in the revised version.

Page 14, Line 26: "Clearly, the model only considering the homogeneous source and without observational constraints predicted much lower levels of the $HO_x$ radicals and AOC at Mt. Tai. Specifically, the discrepancy in the mid-day (9:00-15:00) average $P_{OH}$, OH, $HO_2$, and AOC can be up to 83.4% (63.7%), 47.2% (27.1%), 39.7% (20.3%), and 44.8% (24.9%) in winter (spring), compared to the base scenario with constraints of the measured HONO data."

Page 15, Line 19: "With only inclusion of the OH+NO reactions, significant reductions of the modelled OH (by ~47.2%; 27.1%), $HO_2$ (by ~39.7%; 20.3%), $P_{OH}$ (by ~83.4%; 63.7%), and AOC (by ~44.8%; 24.9 %) were found, compared with being constrained by observed HONO data."

*17. Page 13, Baergen and Donaldson reference – this is a talk not a journal publication. I am fairly sure this group has a journal publication about this.*

**Response:** Thanks for the suggestion. We have found the journal publication of Baergen and Donaldson et al. in 2016, and cited it in the revised manuscript.

"Baergen, A. M. and Donaldson, D. J.: Formation of reactive nitrogen oxides from urban grime photochemistry, Atmos. Chem. Phys., 16, 6355–6363, 2016."

---

## Author Comment (AC2) · 18 Aug 2020

**Response to Reviewer 2:**

*This manuscript describes measurements of Nitrous Acid (HONO) and a suite of other trace gases at a mountaintop site (Mt. Tam Observatory) on the North China Plain. Their goal was to assess whether currently understood HONO sources could explain measured HONO mixing ratios at the (sometimes), free tropospheric site. The authors use their observations coupled with model output to calculate midday HONO photostationary state values (PSS) and compare their observations with Master Chemical Mechanism (MCM) model output. Closing the HONO budget has been a challenging long-standing issue within the community and this paper is a further attempt to do so. The publication is generally well written and the content falls within the scope of Atmospheric Chemistry and Physics. I recommend publication once the following comments are addressed.*

**Response:** We appreciate the reviewer for the positive comments and helpful suggestions. We have carefully considered all of the comments, and revised the original manuscript accordingly. Below we provide the original referee's comments in *black italics*, followed by our responses and changes in the manuscript in blue and red, respectively.

**Major Comments:**

*While the Long Path Absorption Photometer Technique (LOPAP) has been extensively tested for a variety of interferences (Heland et al., 2001; Kleffmann and Wiesen et al., 2008), it has been shown to have an interference from peroxynitric acid ($HO_2NO_2$, PNA), (Legrand et al., 2014). While in many studies ambient temperatures would have rendered this potential interference negligible, I cannot help but feel this is the type of scenario where it could potentially be a problem. The temperatures under which these measurements were performed (and $NO_x$ levels) should support a long enough PNA lifetime for it to be observed. In addition, the authors indicate that the missing HONO source should scale with solar radiation and $NO_2$. PNA should also scale with both of these parameters. While it would be nice to have a full interference test of the instrument for PNA, synthesizing or developing photolytic generation sources can be a challenge. The authors should address this by adding a section discussing the possibility of the interference. Using their MCM model runs they could provide an estimate of what best/worst case scenarios would look like. I doubt that PNA alone is the missing "HONO" source, however constraining the possibility of interference could give a more accurate picture to how much HONO is in fact missing.*

**Response:** We appreciate the reviewer for the constructive comment that we did not consider carefully before. According to the suggestions, we reviewed more literatures and performed more MCM simulations to carefully explore the potential interferences from $HO_2NO_2$ to our HONO measurements. Below are some detailed results and our thoughts about this issue.

(1) We simulated the time series of $HO_2NO_2$ at Mt. Tai in winter and spring by the MCM chemical box model constrained with all our measured species including HONO. As shown in Figure R2-1, moderate concentration levels of $HO_2NO_2$ were predicted by the model at Mt. Tai, with average values (±SD) of 0.07±0.06 ppbv and 0.03±0.04 ppbv in winter and spring, respectively. If we took the $HO_2NO_2$ interference of 15% (Normal Case) reported by Legrand et al. (2014), the potential interference to the excess HONO (measured HONO minus the PSS HONO) were 16%±15% and 11%±10% in winter and spring, respectively. If we assumed a 100% interference (Worst Case), the potential interference from $HO_2NO_2$ to the excess HONO were 72%±30% and 66%±54% in winter and spring.

[Figure]

**Figure R2-1.** Model simulated $HO_2NO_2$ concentrations (grey) and comparison with the measured HONO (red) and excess HONO (green) at Mt. Tai in winter and spring.

(2) We examined the relationship between excess HONO (the measured HONO minus the PSS HONO) and $[NO_2]*J(NO_2)$, and the results are shown in Fig. R2-2. As we can see, the correlation was overall rather weak between excess HONO and $[NO_2]*J(NO_2)$, especially in winter. In comparison, the correlations were improved after the aerosol surface area was taken into consideration, with r of 0.54 and 0.48 between excess HONO and $[NO_2]*J(NO_2)*(S/V)a$ in winter and spring (see Fig. 6 in original manuscript). This indicates that the interference from $HO_2NO_2$ may not be a major factor in the determined excess HONO, and aerosol surface should play an important role.

[Figure]

[Figure]

**Figure R2-2.** Scatter plots of excess HONO versus $[NO_2]*J(NO_2)$ at Mt. Tai in (a) winter and (b) spring.

(3) Based on the literature review, $HO_2NO_2$ is indeed a considerable interference to the LOPAP HONO measurements in the low temperature environments, such as polar regions and high mountain areas. Nonetheless, current studies haven't reached a consensus on the magnitude of $HO_2NO_2$ interference to the LOPAP measurements. For example, Legrand et al. (2014) reported that the $HO_2NO_2$ interference measured by LOPAP was about 15% in their lab experiments. Rappenglück et al. (2014) suggested that the interference could be higher than 15% from their wintertime measurements in an oil and gas field under severe $O_3$ pollution. Kerbrat et al. (2012) reported that an unpublished work by Ammann showed the $HO_2NO_2$ interference to the LOPAP was less than 3%. Obviously, more experiments are still needed to quantify the potential interference from $HO_2NO_2$ to the LOPAP HONO measurements.

In the revised manuscript, we have added a section (see below) to discuss the possible interference from $HO_2NO_2$ to the measured HONO in the present study.

**"Possibility of measurement interference from peroxynitric acid (PNA; $HO_2NO_2$):** While the LOPAP instrument has been extensively tested for a variety of interferences (Heland et al., 2001; Kleffmann and Wiesen, 2008), some recent studies reported that it may be subject to positive interference from $HO_2NO_2$ (e.g., Legrand et al., 2014). Due to the thermo decomposition nature of $HO_2NO_2$, its interference is generally negligible at ambient temperatures at the ground level, but may become important in the circumstances with low temperature and active photochemistry. Legrand et al. (2014) reported that the interference from $HO_2NO_2$ to their HONO measurements was about 15% according to laboratory experiments. In the present study, we did not conduct in-situ measurements of $HO_2NO_2$. To estimate the potential interference for our HONO measurements, we simulated the $HO_2NO_2$ concentrations at Mt. Tai in both campaigns by the MCM chemical box model constrained with all measured species including HONO. Figure S6 shows the time series of modelled $HO_2NO_2$ and its comparison with the measured HONO and missing HONO (measured HONO minus

[HONO]$_{pss}$) concentrations. Overall, moderate concentration levels of $HO_2NO_2$ were predicted by the model at Mt. Tai, with average values (±SD) of 0.07±0.06 ppbv and 0.03±0.04 ppbv in winter and spring, respectively. If we took the $HO_2NO_2$ interference of 15% (Legrand et al., 2014), the potential interference to the missing HONO were 16%±15% and 11%±10% in winter and spring, respectively. Figure S7 shows the scatter plots of missing HONO versus [$NO_2$]*J($NO_2$), an indicator of the $HO_2NO_2$ production. As we can see, the correlation was rather weak between missing HONO and [$NO_2$]*J($NO_2$), especially in winter (r=0.19). This indicates that the interference from $HO_2NO_2$ may not be a major factor in the determined missing HONO, and more experiments are needed to confirm and quantify the possible interferences to the ambient HONO observations."

Kleffmann, J., and Wiesen, P.: Technical Note: Quantification of interferences of wet chemical HONO LOPAP measurements under simulated polar conditions, Atmos. Chem. Phys., 8, 6813-6822, 2008.

Legrand, M., Preunkert, S., Frey, M., Bartels-Rausch, T., Kukui, A., King, M. D., Savarino, J., Kerbrat, M., and Jourdain, B.: Large mixing ratios of atmospheric nitrous acid (HONO) at Concordia (East Antarctic Plateau) in summer: a strong source from surface snow?, Atmos. Chem. Phys., 14, 9963-9976, 2014.

Rappenglück, B., Ackermann, L., Alvarez, S., Golovko, J., Buhr, M., Field, R. A., Soltis, J., Montague, D. C., Hauze, B., Adamson, S., Risch, D., Wilkerson, G., Bush, D., Stoeckenius, T., and Keslar, C.: Strong wintertime ozone events in the Upper Green River basin, Wyoming, Atmos. Chem. Phys., 14, 4909-4934, 2014.

Kerbrat, M., Legrand, M., Preunkert, S., Gallee, H., and Kleffmann, J.: Nitrous acid at Concordia (inland site) and Dumont d'Urville (coastal site), East Antarctica, J. Geophys. Res. Atmos., 117, 2012.

***Specific Comments***
*1. Please indicate where the data and model output are available to readers. Title: Should be "The North China Plain"?*

**Response:** The measurement data and model output used in the present study have been deposited in the Mendeley Dataset, which can be accessed from the following weblink (https://data.mendeley.com/datasets/wcn84cybx9/draft#folder-defadc56-944c-4f33-af54-14019d73ac61). The title has been changed as suggested.

"Data availability. The measurement data and model output used in the present study can be accessed from https://data.mendeley.com/datasets/wcn84cybx9/draft#folder-defadc56-944c-4f33-af54-14019d73ac61. The code for the MCM model can be downloaded from the MCM

website (http://mcm.leeds.ac.uk/MCMv3.3.1/home.htt)".

Title: "Sources of nitrous acid (HONO) in the upper boundary layer and lower free troposphere of the North China Plain: insights from the Mount Tai Observatory"

*2. P1 L20: Are the mixing ratios means or medians? What are they? I believe it is mentioned further in the text, but it should be noted here as well.*

**Response:** They are mean mixing ratios. For clarity, the statement has been revised as follows in the revised version.

"HONO showed moderate concentration levels (average ± standard deviation: 0.15±0.15 and 0.13±0.15 ppbv), with maximum values of 1.14 and 3.23 ppbv in winter and spring, respectively."

*3. P1 L21: Could the noontime max be biased by measuring $HO_2NO_2$ (see major comments)?*

**Response:** As discussed in the "Response to Major Comments", we estimated the potential interferences from $HO_2NO_2$ to our HONO measurements in both Normal and Worst cases, and the relatively weak correlation between missing HONO and $[NO_2]*J(NO_2)$ suggested that the bias by the $HO_2NO_2$ interference should be small.

*4. P1 L27: Would a source consistent with $NO_2$ not also be consistent with $HO_2NO_2$ (see major comments)?*

**Response:** As discussed in the "Response to Major Comments", the correlations between missing HONO and $[NO_2]*J(NO_2)$ were rather weak, especially in winter. In comparison, the correlations were improved after the aerosol surface area was taken into consideration, with r of 0.54 and 0.48 between excess HONO and $[NO_2]*J(NO_2)*(S/V)a$ in winter and spring. This suggests that the $HO_2NO_2$ interference should not be a major factor in the determined missing HONO, and aerosol surface may play an important role.

*5. P4 L9: Change the wording of "Observatory has been widely deployed". You don't really deploy a building. Perhaps "Observatory has been widely used as a sampling location"?*

**Response:** This statement has been revised as suggested in the revised version.

"The Mt. Tai Observatory has been widely used as a sampling location to investigate the regional air pollution in the North China Plain region in the past decade (e.g., Gao et al., 2005;

Sun et al., 2016; Wen et al., 2018)."

*6. P6 L22: What do the authors mean by approximated by the CO (Temperature) data? Was it approximated by CO or temperature? Please clarify.*

**Response:** We are sorry that the original descriptions may be too simplified and unclear. In the present study, the measurements of VOC and carbonyls were made offline by taking air samples followed by laboratory chemical analysis. The VOC and carbonyl samples were only collected during the daytime (7:00-19:00 local time) on some days, and there were no data available for the nighttime period. To facilitate the model simulations, these measured VOC and carbonyl data were approximated to a high resolution (i.e., 5 min) time series as follows. During the daytime when multiple VOC and carbonyl samples were available, the measurement data were directly interpolated to a time resolution of 5 min. For the period when measurement data were unavailable, the VOC concentrations (except for isoprene) were estimated with the real-time CO data by assuming a linear regression relationship between VOCs and CO (the regression was established with the concurrent measurement data of VOCs and CO). The same method was also applied for isoprene and carbonyls, but ambient temperature was used instead of CO for isoprene, and multi-linear regression with CO and $O_3$ was used for carbonyls to account for the primary and secondary sources of carbonyls. For clarify, the following descriptions have been provided in the revised manuscript.

"For VOCs and carbonyls for which the measurements were not in real-time, the high-resolution data were approximated as follows. During the daytime when multiple VOC and carbonyl samples were available, the measurement data were interpolated to a time resolution of 5 min. For the period when measurement data were unavailable, the VOC concentrations (except for isoprene) were estimated with the real-time CO data by assuming a linear regression relationship between VOCs and CO (note that the regression was established with the available measurement data of VOCs and CO). The same method was applied for isoprene and carbonyls, but ambient temperature was used instead of CO for isoprene, and multi-linear regression with CO and $O_3$ was used for carbonyls to account for the primary and secondary sources of carbonyls (Yang et al., 2018; Xue et al., 2016). Such approximation may be subject to some uncertainties but should not significantly interfere the estimation of the role of HONO photolysis in OH sources (Yang et al., 2018)."

*7. P7 L27: I find the description of AOC to be confusing. Are the authors simply trying to describe OH-Reactivity or something else? Please clarify.*

**Response:** We are sorry that the original description is misleading. It is not the OH reactivity, but is defined as sum$\{k_{OH}[OH][X_i]\}$. For clarity, the original statements have been modified

as follows in the revised manuscript.

"Also calculated by the model was the AOC by OH, which is defined here as the reaction rate of OH with NO, $NO_2$, $SO_2$, CO and VOCs (AOC = $\Sigma(k_{OH}[OH][Xi])$: [Xi] is the concentration of the individual reactant species, and $K_{OH}$ is the rate coefficient of OH with Xi)."

*8. P7 L8: This wording is a bit ackward "The atmospheric conditions at Mt. Tai were featured by a cold and dry weather". Perhaps "The atmospheric conditions at Mt. Tai were dominated by cold and dry weather."?*

**Response:** Changed as suggested.

"The atmospheric conditions at Mt. Tai were dominated by cold and dry weather (especially in winter with average (±standard deviation; SD) temperature and RH of -5.2±3.8 °C and 48±20%) as well as relatively lower concentrations of air pollutants."

*9. P9 L18: "It can be argued that the heterogeneous formation of HONO should be stronger at the mountaintop, which may be due to the more intense solar radiation at the high altitudes." Why can this be argued? The ratios are almost within one standard deviation of each other. This would potentially be true assuming the source scales with light intensity, but what would the difference in surface area for the heterogeneous reaction be? That's not clear.*

**Response:** We agree with the reviewer that this statement is a little bit arbitrary. In the revised manuscript, this statement has been revised as follows.

"Third, although the HONO/$NO_2$ ratios were almost comparable at both surface and Mt. Tai during the night, the daytime ratios were significantly higher at the mountaintop (0.065±0.093 and 0.093±0.094) than at the surface (0.047±0.090 and 0.052±0.040). It implies the enhanced HONO formation from the $NO_2$-involved sources at the mountaintop, especially in spring (see Fig. 3)."

*10. P9 L31: Is there no newer reference than 1973? Surely there have been some advances.*

**Response:** A new and more recent reference has been cited in the revised version.

"Around noontime, the PBL has been developed and $K_z$ is generally in the order of $10^6$ $cm^2$ $s^{-1}$ (Zhang et al., 2009)."

"Zhang, N., Zhou, X., Shepson, P. B., Gao, H., Alaghmand, M., and Stirm, B.: Aircraft

measurement of HONO vertical profiles over a forested region, Geophys. Res. Lett., 36, ,172-173, 2009."

*11. P10 L6: The surface HONO would most certainly be extensively diluted by the time it reached the elevated site. Could the authors not use a conserved tracer to estimate the dilution? Or for that matter the upslope time?*

**Response:** As we did not conduct the measurements concurrently both at surface and on the mountaintop, it was indeed difficult for us to find a conserved tracer to accurately estimate the dilution during the upslope transport. Here we only estimated the maximum transport height driven by the turbulent diffusion and mountain-valley breeze, to prove the potential important role of upslope transport in the observed daytime HONO at Mt. Tai.

*12. P13 L20: How do you know the model underestimated the $HO_x$ radical levels? I am unaware of $HO_x$ measurements as part of this campaign constraining this. They aren't noted anywhere. This (as well as in the conclusion) should be re-worded.*

**Response:** The original statements are misleading. We didn't have in-situ HOx measurements in this study, and the inter-comparison was only made between two model scenarios with and without the measured HONO constraints. For clarity, these statements have been revised as follows in the revised version.

Page 14, Line 26: "Clearly, the model only considering the homogeneous source and without observational constraints predicted much lower levels of the $HO_x$ radicals and AOC at Mt. Tai. Specifically, the discrepancy in the mid-day (9:00-15:00) average $P_{OH}$, OH, $HO_2$, and AOC can be up to 83.4% (63.7%), 47.2% (27.1%), 39.7% (20.3%), and 44.8% (24.9%) in winter (spring), compared to the base scenario with constraints of the measured HONO data."

Page 15, Line 19: "With only inclusion of the OH+NO reactions, significant reductions of the modelled OH (by ~47.2%; 27.1%), $HO_2$ (by ~39.7%; 20.3%), $P_{OH}$ (by ~83.4%; 63.7%), and AOC (by ~44.8%; 24.9 %) were found, compared with being constrained by observed HONO data."

*13. Figure 2: While the shaded areas currently represent standard error of the measurement, perhaps showing standard deviation of the measurement might be more useful so that the reader can get a better feel for the range of values observed.*

**Response:** The suggestion has been adopted in the revised manuscript. For a better clarity of the figure, half standard deviation of the measurement data was used in the revised figure, see

below.

[Figure]

Revised Figure 2. Average diurnal variations of (a) HONO, (b) $NO_2$, (c) HONO/$NO_2$, (d) $O_3$, (e) $PM_{2.5}$, (f) CO, (g) $NO_y$, (h) RH, and (i) temperature in winter (red) and spring (black) at Mt. Tai. Shaded area indicates half standard deviation of the measurement data.

*14. Figure 5: A figure showing the relative contribution of each of known HONO formation pathways would also be nice so that readers can get a visual idea of the importance of each pathway at this location.*

**Response:** Thanks for your helpful suggestion. The following figure showing the relative contributions of PSS HONO and unknown HONO formation pathways has been provided in the supporting information in the revised version.

[Figure]

**Figure S4.** Relative contributions of HONO$_{PSS}$ and unknown HONO formation to the observed HONO around noon in winter (left) and spring (right).

---

## Author Comment (AC3) · 18 Aug 2020

**Response to Reviewer 3:**

*The work by Jiang et al presents measurements of HONO and other supporting species at Mt Tai, a mountaintop site in the North China Plain. Concurrent ground level measurements were also performed, and comparison were made to explore the source differences. The authors present an interesting data set and provide modelling work to aid their interpretation. Overall, this paper is well thought out and written, with the results clearly presented in the tables and figures. I would recommend publication after consideration to the comments below.*

**Response:** We appreciate the reviewer for the positive comments and helpful suggestions. We have carefully considered all of the comments, and revised the original manuscript accordingly. Below we list the original referee's comments in black *italics*, followed by our responses and changes in the manuscript shown in blue and red, respectively.

**Minor comments**

*1. Section 2.2: Were the same instruments used at Mt. Tai as the ground level monitoring stations? This is particularly important for HONO measurements, as previous work has shown significant differences can be reported for co-located HONO instruments, even of the same type (See e.g. Crilley et al., 2019). Furthermore, were there any inter-comparison measurements of the instruments at Mt Tai and ground to account for any differences between instruments that may affect the comparison?*

**Response:** We are sorry that the original description may be misleading. The same LOPAP instrument was used at both Mt. Tai and the ground-level site (Ji'nan). Please note that both measurement campaigns were not concurrent, and the 1-yr continuous HONO measurements in Ji'nan were carried out from September 2015 to August 2016, earlier than the present study at Mt. Tai (2017 winter and 2018 spring).The measurement protocol and quality assurance and quality control procedures of LOPAP were also the same between the two measurement campaigns. For clarity, the following statements have been provided in the revised manuscript.

"It should be noted that a 1-yr continuous measurement campaign had been conducted from September 2015 to August 2016 at an urban site of Ji'nan using the the same instrument (Li et al., 2018a), and their results are compared here with those at Mt. Tai to infer the vertical distributions of HONO in the NCP region."

*2. Page 5, line 3: Typically, the baseline is measured every 4-8 hours with a LOPAP (see e.g. Crilley et al., 2019; Kleffmann and Weisen 2008) to capture the temporal variability. Measuring the baseline every 11h 30 min may not be sufficient to capture the baseline variability and I am curious why the authors chose to do it like this.*

**Response:** Thanks for the helpful suggestion. Considering the relatively clean conditions at such a high elevation station, we chose to perform the baseline correction in a relatively longer time interval (i.e., 11 h 30 min) in this study. We further examined the temporal variability of the measured baseline during this study, and it was quite stable (see figure below). We believe that this choice should not affect the quality of observations presented here, and will take this into consideration in our future studies with different pollution conditions.

[Figure]

**Figure R3-1.** Raw signal of LOPAP instrument showing the baseline calibration (grey shaded).

*3. Page 6, line 22: I don't quite follow this sentence 'based on the CO (temperature) data and the measured correlations with CO (temperature) for anthropogenic (biogenic) VOCs' What do you mean by CO (temperature)?*

**Response:** We are sorry that the original descriptions may be too simplified and unclear. We have elaborated more about it in the revised manuscript by the following discussions.

"For VOCs and carbonyls for which the measurements were not in real-time, the high-resolution data were approximated as follows. During the daytime when multiple VOC and carbonyl samples were available, the measurement data were interpolated to a time resolution of 5 min. For the period when measurement data were unavailable, the VOC concentrations (except for isoprene) were estimated with the real-time CO data by assuming a linear regression relationship between VOCs and CO (note that the regression was established with the available measurement data of VOCs and CO). The same method was applied for isoprene and carbonyls, but ambient temperature was used instead of CO for isoprene, and multi-linear regression with CO and $O_3$ was used for carbonyls to account for the primary and secondary sources of carbonyls (Yang et al., 2018; Xue et al., 2016). Such approximation may be subject to some uncertainties but should not significantly interfere the estimation of the role of HONO photolysis in OH sources (Yang et al., 2018)."

*4. Page 10, line 23: are the reported j(HONO) and OH concentrations noontime maxima or daily average?*

**Response:** They are the average values at noontime (11:00-14:00 LT). The original statement has been revised as follows for clarity.

"According to the measurement-derived J(HONO) (with noontime averages of 6.4±3.5 and 9.5±3.2×10$^{-4}$ s$^{-1}$ in winter and spring; see Fig. S3) and the model-simulated OH concentrations (with noontime averages of 2.5±0.7 and 4.4±2.0×10$^6$ molecules cm$^{-3}$; Fig. S3), the average lifetime of HONO was estimated as 25.7±1.4 and 21.8±16.9 minutes during noontime (11:00-14:00 LT) in winter and spring, respectively."

*5. Page12, line7: Here you state that heterogenous reaction with NO$_2$ on aerosol surfaces should be a significant daytime HONO source at Mt. Tai rather than on ground. What is ratio of ground vs aerosol surface area? I am asking to try and understand why this NO$_2$ reaction may preferentially occur on aerosol surfaces at Mt Tai, unlike previous work at ground level.*

**Response:** Actually, we cannot make an accurate estimation for the ratio of ground versus aerosol surface areas. Here we argue that aerosol surface may play a more important role than ground surface in the heterogeneous formation of HONO mainly due to the following reasons. First, Mt. Tai (1534 m a.s.l.) is the highest mountain over the North China Plain, and the station is situated on an isolated peak of the mountain. Thus, the measurement site is far away from the ground level surface, and the terrestrial surface is much limited compared to the ground level studies. Second, the correlation analysis showed much stronger correlation between P$_{other}$ and NO$_2$*(S/V)a (the commonly used indicator for the heterogenous HONO formation on aerosol surface) than that between P$_{other}$ and NO$_2$ (the indicator for heterogeneous HONO formation on ground surface). More measurements are still needed to better understand the potential role of aerosol surface in the HONO formation in the high elevation atmospheres.

*6. Page 13, line 10: it would also be good to report the percentage HONO photolysis contributes to OH production from the model, as this would enable comparison to other work.*

**Response:** Thanks for the helpful suggestion. The following statements have been added in the revised manuscript.

"In percentage, HONO photolysis accounted for 44.4% and 25.8% of the total primary RO$_x$ production at mid-day at Mt. Tai in winter and spring, respectively. For OH alone, the percentages of the contribution of HONO photolysis to the primary sources were 93.2% and 71.3% in winter and spring."

*7. Figure 3: Why is there so much more noise in the HONO/NO₂ diurnal plots compared to HONO and NO₂?*

**Response:** We examined the measured time series of HONO, NO₂ and HONO/NO₂ ratios, and indeed found larger fluctuations in the HONO/NO₂ ratio than those of HONO and NO₂. The fluctuation in HONO/NO₂ was amplified by the variability of NO₂ concentrations, with much higher HONO/NO₂ values at lower NO₂ levels. So, the large fluctuations in HONO/NO₂ should be partly due to the relatively lower levels of NO₂ at Mt. Tai. In comparison with the ground-level site in urban Ji'nan, the fluctuation in the measured HONO/NO₂ was much smaller given its much higher levels of NO₂.

*8. Figure 5: I am surprised that there is no noon-time maxima in HONO$_{pss}$, as the OH should peak then (as seen in Fig 8) along with the NO$_x$? (as shown in Fig 2)?*

**Response:** We are sorry that Figure 5 with a log coordinate for y-axis was not clear to show the noon-maxima in HONO$_{pss}$. As shown from the figure below, HONOpss actually showed a noontime maximum, and the diurnal pattern of HONO$_{pss}$ followed well with that of [NO]*[OH].

[Figure]

**Figure R3-2.** Average mixing ratios of NO, OH, HONO$_{pss}$ and [NO]*[OH] concentrations around noon (11:00-14:00 LT) in winter (a and c) and spring (b and d).